# Multidecadal Polynya Formation in a Conceptual (Box) Model

Daan Boot[1], René M. van Westen[1], and Henk A. Dijkstra[1,2]

[1]Institute for Marine and Atmospheric research Utrecht, Department of Physics, Utrecht University, Utrecht, the Netherlands
[2]Center for Complex Systems Studies, Utrecht University, Utrecht, the Netherlands

*Correspondence to:* D. Boot <d.boot@uu.nl>

**Abstract.** Maud Rise Polynyas (MRPs) form due to deep convection, which is caused by static instabilities of the water column. Recent studies with the Community Earth System Model (CESM) have indicated that a multidecadal varying heat accumulation in the subsurface layer occurs prior to MRP formation due to the heat transport over the Weddell gyre. In this study, a conceptual MRP box model, forced with CESM data, is used to investigate the role of this subsurface heat accumulation in MRP formation. Cases excluding and including multidecadal varying subsurface heat and salt fluxes are considered and multiple polynya events are only simulated in the cases where subsurface fluxes are included. The dominant frequency for MRP events in these results, approximately the frequency of the subsurface heat and salt accumulation, is still visible in cases where white noise is added to the freshwater flux. This indicates the importance and dominance of the subsurface heat accumulation in MRP formation.

## 1 Introduction

The Weddell Sea is a region where open-ocean polynyas occasionally form. A distinction is made between the larger Weddell Sea Polynyas (WSPs), and the smaller Maud Rise Polynyas (MRPs). Formation of the MRPs is clearly related to bathymetry, i.e. Maud Rise, an underwater seamount, while this clear relation is absent for WSPs. The first polynyas in the Weddell Sea were observed in the 1970s around 65°S and 0°E (Martinson et al., 1981) with in-situ observations (Gordon, 1978) and first available satellite images (Carsey, 1980). In 1974, 1975, and 1976 polynyas with an areal extent of approximately $2.5 \times 10^5$ km$^2$ were present during the entire winter (Gordon et al., 2007) and can be classified as WSPs. An MRP appeared in 2016 and 2017 (Jena et al., 2019). The 2017 MRP had an approximate area of $0.5 \times 10^5$ km$^2$ and persisted from September to October (Campbell et al., 2019; Cheon and Gordon, 2019). Observations also suggest a short-lived and small-scale MRP in 1994 (Holland, 2001; Lindsay et al., 2004). In this paper we will focus specifically on MRPs.

Many studies have looked into the processes responsible for MRP formation. A key theory is that the MRP is formed by deep convection caused by static instability due to surface salt anomalies in a preconditioned water column (Martinson et al., 1981). Such salt anomalies can be caused by brine rejection (Martinson et al., 1981) but also by freshwater flux anomalies due to variations in the Southern Annular Mode (Gordon et al., 2007; Cheon and Gordon, 2019). Another line of work focuses on dynamical forcing by the wind (Parkinson, 1983; Francis et al., 2019; Jena et al., 2019; Campbell et al., 2019) as a cause of MRP formation. A divergent wind stress can open up the sea-ice pack and induce upwelling (by Ekman dynamics) which either causes an MRP directly or induces deep convection. Finally, dynamical forcing through ocean eddy shedding at the flanks of Maud Rise (Holland, 2001) and Taylor caps above Maud Rise (Alverson and Owens, 1996; Kurtakoti et al., 2018) can change

the background stratification and hence precondition the water column. These processes also cause a general halo of relatively low sea-ice concentrations over Maud Rise (Lindsay et al., 2004).

Climate models provide the opportunity to study deep convection and consequently MRP formation. From several climate models, it is known that deep convection in the Southern Ocean varies on multidecadal to multicentennial time scales (Martin et al., 2013; Zanowski et al., 2015; Latif et al., 2017; Weijer et al., 2017). Several climate models show subsurface heat accumulation prior to deep convection, e.g. in the Kiel Climate Model (KCM) (Martin et al., 2013), the Community Earth System Model (CESM) (van Westen and Dijkstra, 2020a) and the Geophysical Fluid Dynamics Laboratory Climate Model (GFDL CM2-0) (Dufour et al., 2017). Through buoyancy gain in the subsurface layer, deep convection is induced, which results in MRP formation (Martin et al., 2013; Latif et al., 2017; Reintges et al., 2017). In the KCM, for example, a stronger stratification results in a longer period for deep convection, because more buoyancy gain is necessary to overcome the more stable stratification (Latif et al., 2017; Reintges et al., 2017). Another important feature is model resolution as shown by Weijer et al. (2017): MRPs were found in a high-resolution ($0.1°$) version of the CESM, whereas in the low-resolution ($1°$) version of the same model no MRPs were simulated. Dufour et al. (2017) used the GFDL CM2-0 model with a nominal ocean grid spacing of $0.25°$ and $0.1°$ and they show that the occurrence of deep convection itself is not sufficient to create MRPs. If the subsurface heat reservoir cannot supply enough heat to melt all the sea ice, an MRP will not form.

A recent model study by van Westen and Dijkstra (2020a) shows a multidecadal occurrence of MRPs and suggest that the time scale of MRP formation is affected by intrinsic ocean variability through subsurface preconditioning. They relate the subsurface heat accumulation near Maud Rise to the Southern Ocean Mode (SOM), a multidecadal mode of intrinsic variability in the Southern Ocean caused by eddy-mean flow interactions (Le Bars et al., 2016; Jüling et al., 2018), which is present in high-resolution (ocean) models (van Westen and Dijkstra, 2017). van Westen and Dijkstra (2020a) show that heat content anomalies propagate from the SOM region ($50°S – 35°S \times 50°W – 0°W$) via the Antarctic Circumpolar Current (ACC) to $30°E$, and the Weddell Gyre to the Maud Rise area where they cause heat accumulation in the subsurface layer. The CESM model results are further analysed in van Westen and Dijkstra (2020b) where the importance of this subsurface heat accumulation on the MRP formation is shown.

To better understand the results of the high-resolution CESM simulations (van Westen and Dijkstra, 2020a, b), and to connect to earlier theories of MRP formation, we use here an extension of the Martinson et al. (1981) box model. This extended version of the Martinson model is described in Section 2. Results for five different cases are considered (Section 3), to address the importance of subsurface forcing (i.e. heat and salt accumulation) relative to surface forcing (e.g., brine rejection, wind forcing) in MRP formation, and the processes determining the long-term variability of MRPs. In Section 4, a summary and discussion of the results is given.

## 2  Model description and cases considered

The model used is based on the box model proposed in Martinson et al. (1981). Note that this box model was originally developed for Weddell Sea Polynyas (WSP). Martinson et al. (1981) proposed that convection was initiated by surface salin-

ity anomalies, similar to the surface-initiated convection processes suggested during MRP formation (Kurtakoti et al., 2018; Campbell et al., 2019; Cheon and Gordon, 2019; Kaufman et al., 2020). Stratified Taylor columns contribute to the preconditioning of the Maud Rise region by lowering the stratification over Maud Rise compared to the surroundings (Alverson and Owens, 1996; de Steur et al., 2007; van Westen and Dijkstra, 2020a). Nonetheless, a distinct surface and subsurface layer is
present over Maud Rise, which can be simplified by two boxes as is done in Martinson et al. (1981). The original box model is applicable to the Maud Rise with some adjustments. This new model is described in Section 2.1, the CESM simulation used in this study is shortly described in Section 2.2, and the different configurations and cases for the box model are presented in Section 2.3.

## 2.1 Model Description

The MRP box model consists of two vertically stacked boxes with a constant depth and within each box the ocean properties (e.g. temperature and salinity) are uniform. The model simulates the development of temperature ($T$), salinity ($S$), and sea-ice thickness ($\delta$) in each box under surface and subsurface forcing. The depth of the entire water column is $H$ with the surface layer having a depth of $h_1$ and the subsurface layer a depth of $h_2$.

The model has four different flow regimes, below referred to as regimes, which are differentiated on sea-ice cover (sea-ice
free versus sea-ice covered) and static stability (two layered versus mixed). Whenever stable/unstable is mentioned below, we refer to static stability of the water column, so no dynamical instabilities. There are the sea-ice free regimes I and II, and the sea-ice covered regimes III and IV. Regimes II and IV are stably stratified ($\rho_1 < \rho_2$), and regimes I and III are mixed with one uniform density over the entire depth (Fig. 1). The subscripts 1 and 2 correspond to the surface and subsurface layer, respectively.

Over time the model state may transit through these four regimes under the influence of (seasonal) forcing. The four different regime transitions are indicated by the arrows in Fig. 1, i.e.,

(a) From sea-ice covered regimes to sea-ice free regimes due to complete melt of the sea ice ($\delta = 0$) (regime IV $\rightarrow$ II or regime III $\rightarrow$ I),

(b) from sea-ice free regimes to sea-ice covered regimes, because the surface layer reaches freezing temperature and sea ice
starts to form ($T_1 = T_f$) (regime II $\rightarrow$ IV, and regime I $\rightarrow$ III),

(c) from stable, two layered regimes to unstable, mixed regimes, because the density of the surface layer is equal to or larger than that of the subsurface layer ($\rho_1 \geq \rho_2$) (regime II $\rightarrow$ I, and regime IV $\rightarrow$ III). The water column becomes unstable and mixes through overturning. Temperature and salinity are uniform over the entire layer and indicated by $T$ and $S$ instead of $T_n$ and $S_n$, and

(d) from unstable, mixed regimes to stable, two layered regimes, because of stabilisation of the water column due to a decreasing density of the mixed layer (regime I $\rightarrow$ II, and regime III $\rightarrow$ IV).

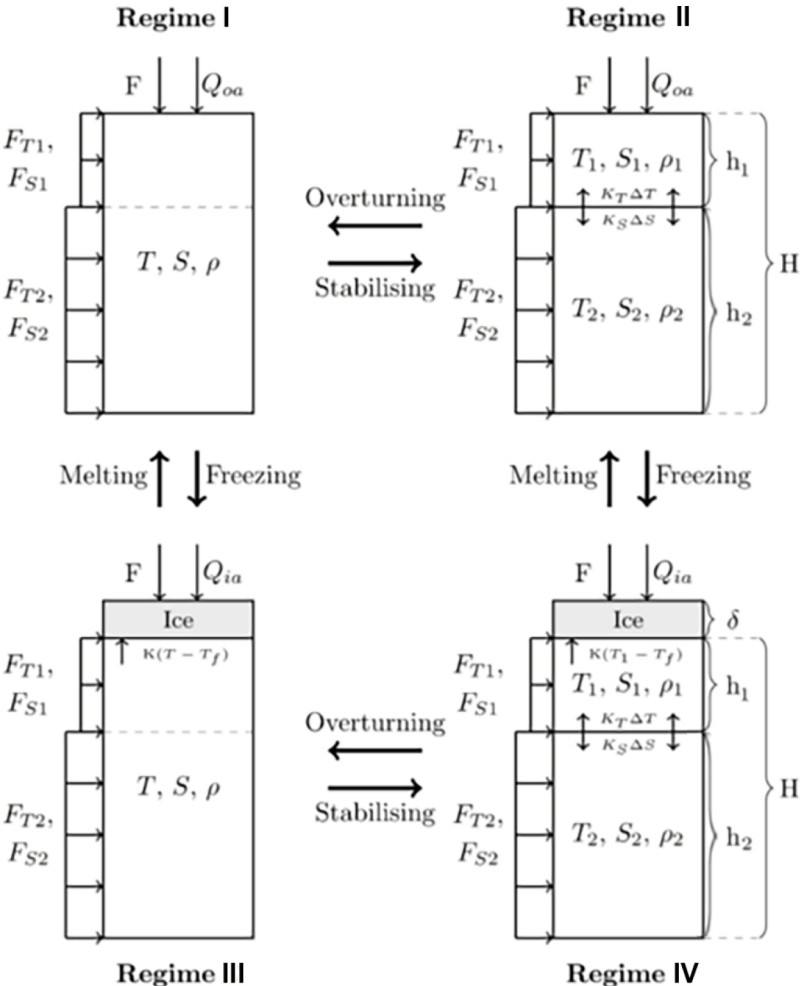

**Figure 1.** A schematic representation of the different regimes of the extension of the model used in Martinson et al. (1981). The parameters displayed in the figure are explained in the text. The directions and size of the arrows are not necessarily a representation of the actual direction and magnitude of the fluxes. The actual size and direction are dependent on the state of the model. Positive fluxes represent fluxes entering the water column. Regime transitions are shown by bold arrows.

The precise conditions for the regime transitions are shown in Appendix A. It should be noted that for the model to switch between regimes I (mixed, sea-ice free) and III (mixed, sea-ice covered) the entire water column should reach freezing temperature, which is physically not realistic. Therefore, this transition does not exist in the model (regime I $\nrightarrow$ regime III)

The model is forced at the surface by a freshwater flux $F$, and by a heat flux $Q_{ia}$ for sea-ice covered regimes and $Q_{oa}$ for open-ocean regimes. The freshwater flux $F$ is modelled as a virtual salt flux by multiplying it with a base salinity $S_0$. Both the surface and subsurface layer are subject to horizontal advective heat and salt fluxes ($F_{T1}$ and $F_{S1}$ for the surface layer, and

$F_{T2}$ and $F_{S2}$ for the subsurface layer) which depend on a background value ($T_{b1}$ and $S_{b1}$ for the surface layer, and $T_{b2}$ and $S_{b2}$ for the subsurface layer) and a relaxation timescale ($\tau$).

The heat and salt transfer between the layers are modelled using exchange coefficients ($K_T$ and $K_S$), which account for upwelling, turbulent exchange and diffusion. In sea-ice covered regimes there is a heat flux present between the sea ice and the underlying layer. This heat flux is modelled using a turbulent exchange coefficient ($K$). Brine is rejected during sea-ice growth and freshwater is added to the surface layer during sea-ice melt, brine rejection and sea-ice melt are modelled using a constant representing the salinity difference between sea ice and seawater ($\sigma$) and the rate of sea-ice growth ($d\delta/dt$). The density for each layer is determined from a simple linear equation of state:

$$\rho = \rho_0(1 - \alpha T + \beta S) \tag{1}$$

where the constants $\alpha$ and $\beta$ are the thermal expansion and haline contraction coefficients, respectively.

The governing equations for each regime are then:

Regime I:

$$H\frac{dT}{dt} = \frac{Q_{oa}}{\rho_0 C_p} + \tau(T_{b1} - T)h_1 + \tau(T_{b2} - T)h_2 \tag{2a}$$

$$H\frac{dS}{dt} = -FS_0 + \tau(S_{b1} - S)h_1 + \tau(S_{b2} - S)h_2 \tag{2b}$$

$$\delta = 0 \tag{2c}$$

Regime II:

$$h_1\frac{dT_1}{dt} = \frac{Q_{oa}}{\rho_0 C_p} + K_T(T_2 - T_1) + \tau(T_{b1} - T_1)h_1 \tag{3a}$$

$$h_1\frac{dS_1}{dt} = K_S(S_2 - S_1) - FS_0 + \tau(S_{b1} - S_1)h_1 \tag{3b}$$

$$\delta = 0 \tag{3c}$$

$$\frac{dT_2}{dt} = \tau(T_{b2} - T_2) + \frac{K_T(T_1 - T_2)}{h_2} \tag{3d}$$

$$\frac{dS_2}{dt} = \tau(S_{b2} - S_2) + \frac{K_S(S_1 - S_2)}{h_2} \tag{3e}$$

Regime III:

$$H\frac{dT}{dt} = K(T - T_f) + \tau(T_{b1} - T)h_1 + \tau(T_{b2} - T)h_2 \tag{4a}$$

$$H\frac{dS}{dt} = \sigma\frac{d\delta}{dt} - FS_0 + \tau(S_{b1} - S)h + \tau(S_{b2} - S)h_2 \tag{4b}$$

$$\frac{d\delta}{dt} = \frac{1}{\rho_i L}(-Q_{ia} - \rho_0 C_p K(T - T_f)) + \frac{FS_0}{\sigma} \tag{4c}$$

Regime IV:

$$h_1\frac{dT_1}{dt} = K_T(T_2 - T_1) - K(T_1 - T_f) + \tau(T_{b1} - T_1)h_1 \tag{5a}$$

$$h_1\frac{dS_1}{dt} = K_S(S_2 - S_1) + \sigma\frac{d\delta}{dt} - FS_0 + \tau(S_{b1} - S_1)h_1 \tag{5b}$$

$$\frac{d\delta}{dt} = \frac{1}{\rho_i L}(-Q_{ia} - \rho_0 C_p K(T_1 - T_f)) + \frac{FS_0}{\sigma} \tag{5c}$$

$$\frac{dT_2}{dt} = \tau(T_{b2} - T_2) + \frac{K_T(T_1 - T_2)}{h_2} \tag{5d}$$

$$\frac{dS_2}{dt} = \tau(S_{b2} - S_2) + \frac{K_S(S_1 - S_2)}{h_2} \tag{5e}$$

In these equations, $C_p$ is the specific heat of seawater with a reference density $\rho_0$. Sea ice has a density of $\rho_i$ and a latent heat of melting/freezing indicated by *L*. The standard values of all parameters used in the model are presented in Section 2.3.

In the temperature equations the ocean-atmosphere and ice-ocean heat flux, horizontal advective fluxes, and heat transfer between the two layers are represented and transformed into temperature changes (units °C/s). The temperature change due to the ocean-atmosphere flux is given by $\frac{Q_{oa}}{\rho_0 C_p h_n}$, where $h_n$ is either $H$ (regimes I and III) or $h_1$ (regimes II and IV). The horizontal advective fluxes result in temperature changes given by $\tau(T_{b1} - T_n)$ (surface) and $\tau(T_{b2} - T_n)$ (subsurface), where $T_n$ is either $T$ (regimes I and III), $T_1$ (surface flux, regimes II and IV) or $T_2$ (subsurface flux, regimes II and IV). The temperature change due to the heat transfer between the sea-ice and the surface layer is represented by $\frac{K(T_n - T_f)}{h_n}$ where $T_n$ is either $T$ or $T_1$, and $h_n$ is either $h_1$ or $H$ depending whether the model is in regime III or regime IV. Lastly, the term representing a temperature change due to heat transfer between the layers in regimes II and IV is given by $\frac{K_T(T_n - T_m)}{h_n}$ where $n$ and $m$ are either 1 or 2.

In the salinity equations the freshwater flux, sea ice melt and brine rejection, horizontal advective fluxes and salt transfer between the two layers are represented and transformed into salinity changes (units g/kg/s). The salinity change due to the freshwater flux is given by $\frac{FS_0}{h_n}$, where $h_n$ is either $H$ or $h_1$. The horizontal advective fluxes result in salinity changes given by $\tau(S_{b1} - S_n)$ (surface) and $\tau(S_{b2} - S_n)$ (subsurface), where $S_n$ is either $S$ (regime I and III), $S_1$ (surface flux, regimes II and IV) or $S_2$ (subsurface flux, regimes II and IV). The salt transfer between the layers in regimes II and IV result in a salinity change given by $\frac{K_S(S_n - S_m)}{h_n}$ where $n$ and $m$ are either 1 or 2. Brine rejection and sea ice melt are given by $\sigma\frac{d\delta}{dt}$.

The sea-ice thickness is dependent on heat transfer between the sea ice and the ocean and the atmosphere, as well as the freshwater flux on top of the ice. Heat transfer between the sea ice and the atmosphere influences the sea-ice thickness via $\frac{Q_{ia}}{\rho_i L}$. Sea ice growth is affected by the heat transfer between the ocean and the sea ice via $\frac{\rho_0 C_p K(T-T_f)}{\rho_i L}$ where $T$ is either $T$ or $T_1$. Sea-ice growth due to precipitation is given by the term $\frac{FS_0}{\sigma}$.

The set of differential Eq. $(2-5)$ is solved using the ODE15s (Ordinary Differential Equation) solver incorporated in Matlab. The ODE15s solver is a variable-step, variable-order solver based on an algorithm by Klopfenstein (1971) using numerical differentiation formulas (NDFs) orders 1 to 5. Tolerances for the absolute and relative error are used to increase the accuracy of the model; these tolerances are set to $10^{-10}$ and $10^{-8}$, respectively.

## 2.2    CESM simulation

In this study we use the results of the same CESM simulation of van Westen and Dijkstra (2020a) to force the MRP Box model. For a full description and details how the simulation was performed we refer to van Westen et al. (2020). The CESM configuration has a horizontal ocean- and sea-ice model resolution of $0.1°$ (10 km). The atmospheric component has a horizontal resolution of $0.5°$ (50 km). The ocean (atmosphere) component has 42 (30) non-equidistant depth (pressure) levels. The model was run for 300 years under year 2000 forcing conditions. For this study, we have used model output from model years $150-$

250, which were detrended as described in van Westen and Dijkstra (2020a).

An example of a polynya in model year 231 of the CESM simulation is shown in Fig. 2. The Maud Rise region is first completely covered with sea ice, though the sea ice is less thick in the Maud Rise region compared to the surrounding (Fig. 2a, b). In model year 231, the polynya forms in mid August (Fig. 2c), after which it extends to a larger polynya in November (Fig. 2d). The analysis in van Westen and Dijkstra (2020b) has indicated that polynya formation in the CESM simulation is not

caused by a bias in the mean state of the CESM (Small et al., 2014) but due to deep convection initiated in the subsurface.

## 2.3    MRP Box Model Setup and Case Description

The original polynya model of Martinson et al. (1981) is obtained from the model formulation above by setting the horizontal advective fluxes ($F_{T1}$, $F_{S1}$, $F_{T2}$ and $F_{S2}$) to zero and by setting the subsurface layer values of temperature and salinity to constant values. We consider two cases of this configuration of the model, which only differ by the value of $K_S$ used. The

higher value of $K_S$ (case MKH: Martinson $K_S$ high) results in more salt transfer from the subsurface layer to the surface layer (than in case MKL (Martinson $K_S$ low) with the lower value of $K_S$) increasing the density of the surface layer, making it more susceptible to overturning (Table 1).

The three cases with a dynamic subsurface layer are differentiated on the inclusion of the different components of the subsurface forcing. Case PFB (Periodic Flux Both) uses both a time varying subsurface heat and salt flux. Case PFH (Periodic

Flux Heat) uses a time varying subsurface heat flux and the subsurface salt flux is set to a constant value. Case PFS (Periodic Flux Salt) uses a time varying salt flux and the heat flux is set to a constant value. The aim of this configuration is to reproduce the general features of the CESM simulation of van Westen and Dijkstra (2020a), where the observed multidecadal variability

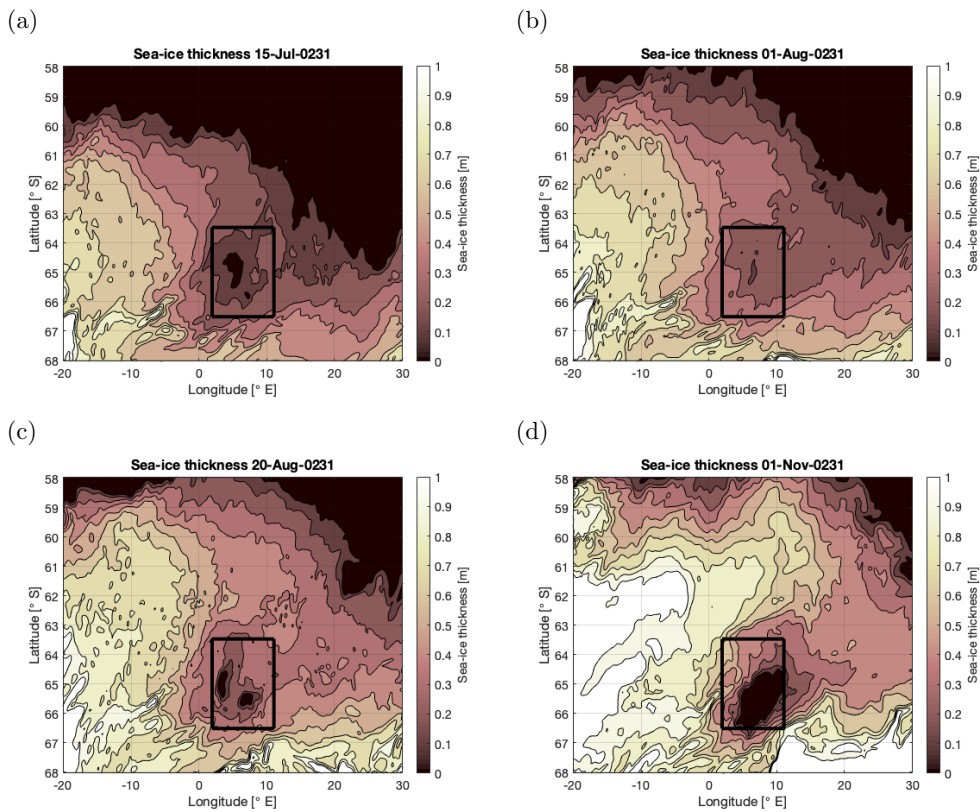

**Figure 2.** Daily-averaged sea-ice thickness for four days during model year 231 when a Maud Rise polynya forms in the CESM simulation. The black outlined region represents the polynya region $2°E - 11°E \times 66.5°S - 63.5°S$ as defined in van Westen and Dijkstra (2020a).

of the MRP events is one of the key features. The different cases are used to assess the importance of the different components of the subsurface forcing on the MRP formation.

Parameter values for each case are displayed in Table 2. The parameter values are either taken from Martinson et al. (1981), or based on the CESM simulation of van Westen and Dijkstra (2020a), or they are determined through tuning of the model.

5   For the CESM simulation, we determined spatial-averaged quantities over the Polynya region ($2°E - 11°E \times 66.5°S - 63.5°S$, Figure 2), where an MRP forms in the CESM (van Westen and Dijkstra, 2020a). The aim of the model is to investigate multiple polynya events, and therefore it is necessary to tune the stratification. The stratification can be either too strong and no overturning occurs, or the stratification is too weak, and the water column overturns each year. To obtain multiple polynya events the heat and salt fluxes between the two layers and between the sea ice and the surface layer are tuned.

10   The typical depth of the layers has been determined from the CESM simulation of van Westen and Dijkstra (2020a). The depth of the surface layer ($h_1$) is set to 160 m, because the potential density shows a clear homogeneous layer below 160 m (Fig. 3a). This compares well to the value used in Kurtakoti et al. (2018) ($h_1 = 150$ m) but is smaller than the value used in

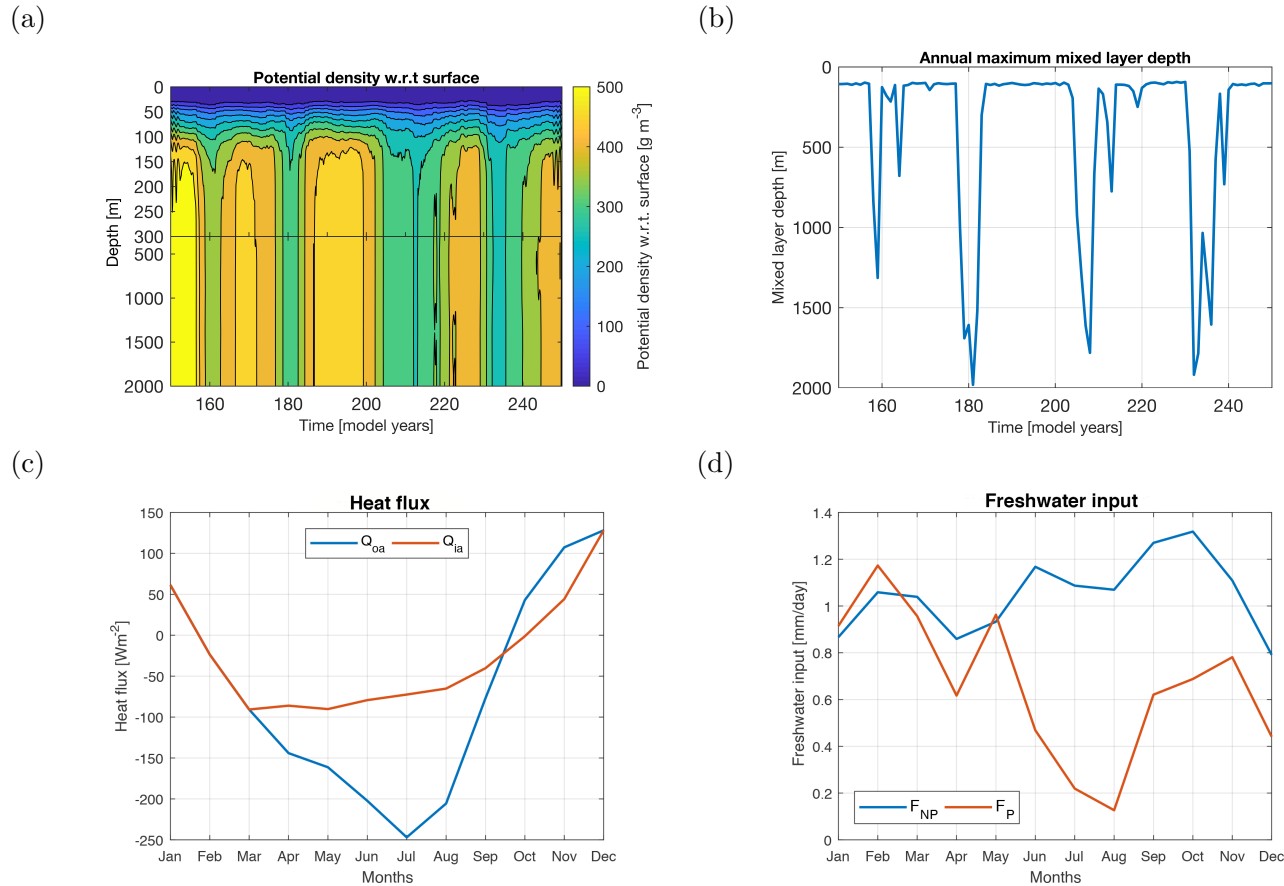

**Figure 3.** (a) Potential density over depth for model years $150 - 250$ from the CESM simulation, the time series are smoothed through a 5-year running mean. (b) The annual maximum mixed layer depth for model years $150 - 250$ from the CESM simulation. (c) The heat fluxes in W m$^{-2}$ (see also Table 3) are determined from the CESM simulation. These values are interpolated linearly in the model as displayed in this figure. (d) The freshwater input in mm day$^{-1}$ (see also Table 3) is determined from the CESM simulation. All quantities are spatially averaged over the Polynya region ($2°E - 11°E \times 66.5°S - 63.5°S$, Figure 2).

**Table 1.** Overview of the different cases considered and values for the diffusivity parameters for heat ($K_T$) and salt ($K_S$) transfer between the two layers for each case. A model component can either be included ('on') or excluded ('off'). The model component 'dynamic $T_2$ and $S_2$' stands for an active subsurface layer. If this component is excluded, the set up uses a subsurface layer with constant density. If either $F_{T2}$ or $F_{S2}$ is excluded, the background value corresponding to the flux is set constant. The model components containing '$F$' represent fluxes with subscripts representing the horizontal heat ($T$) and salt ($S$) fluxes in either the surface (1), or subsurface (2) layer.

| Model Case | Dynamic $T_2$ and $S_2$ | $F_{T1}$, $F_{S1}$ | $F_{T2}$ | $F_{S2}$ | $K_T$ [$10^{-6}$ m s$^{-1}$] | $K_S$ [$10^{-6}$ m s$^{-1}$] |
|---|---|---|---|---|---|---|
| MKL | off | off | off | off | 5.00 | 1.375 |
| MKH | off | off | off | off | 5.00 | 2.00 |
| PFB | on | on | on | on | 3.1 | 3.1 |
| PFH | on | on | on | off | 3.1 | 3.1 |
| PFS | on | on | off | on | 3.1 | 3.1 |

Martinson et al. (1981) ($h_1 = 200$ m). The depth of the entire layer ($H$) is set to 2000 m. This is the approximate mixed layer depth during convective events in the CESM simulation (Fig. 3b). This magnitude corresponds well to values presented in Fahrbach et al. (2011) for the lower limit of where Weddell Sea Deep Water is found, and in Dufour et al. (2017) for the depth of the subsurface layer. It is, however, half the size of the value used in Martinson et al. (1981) ($H = 4000$ m). Cases MKL and
MKH use different subsurface temperature and salinity values. These values are the time-mean temperature and salinity of the subsurface forcing (shown in Section 3.1 and as dashed lines in Fig. 4) in the PFB, PFH and PFS cases. The used values are $T_2 = 1.175$ °C and $S_2 = 34.7847$ g/kg, where Martinson et al. (1981) use $T_2 = 0$°C and $S_2 = 34.66$ g/kg.

The turbulent exchange coefficient $K$, and the exchange coefficients $K_T$ and $K_S$ have been used as tuning parameters for the different cases. The coefficient $K$ is set to $1 \times 10^{-4}$ ms$^{-1}$ for all cases (in Martinson et al. (1981) this value is $3 \times 10^{-4}$ ms$^{-1}$).
The values of $K_T$ and $K_S$ per case are shown in Table 1. To compare the magnitude of these parameters with values used in literature the values need to be converted from ms$^{-1}$ to m$^2$s$^{-1}$, which is the usual unit for vertical diffusivity parameters. This is done by multiplying these values with the depth of the surface layer (i.e. 160 m). This results in values between $2.2 \times 10^{-4}$ m$^2$s$^{-1}$ and $5 \times 10^{-4}$ m$^2$s$^{-1}$. Comparable values are found in a model study of Dufour et al. (2017) for this same location and in observations (Shaw and Stanton, 2014). The values used in this study are up to a factor 10 larger than the values
used in Martinson et al. (1981) ($K_T = 7 \times 10^{-7}$ ms$^{-1}$ and $K_S = 10^{-7}$ ms$^{-1}$).

Initial conditions of the model affect the long-term behaviour of the model. Besides, the initial regime of the model should match with initial conditions. For example, when starting in a sea-ice covered regime, the initial conditions for sea ice should be $\delta > 0$. Another important initial condition is the initial stratification. If the initial stratification is too weak, the model will overturn each year. In this specific case, it is not possible to study multidecadal variability in polynya formation. Therefore,
each model simulation is initiated on January 1 with the following conditions: $T_1 = 0.1$°C, $S_1 = 34.2$ g/kg and $\delta = 1$ m.

**Table 2.** Standard parameter values used in the model. Superscripts show whether the parameter value is determined from the CESM simulation of van Westen and Dijkstra (2020a) (C), or determined through tuning (t), or taken from Martinson et al. (1981) (M). Overbars represent mean values (averaged over a 25-year cycle).

| Parameter | Value | Parameter | Value | Parameter | Value |
|---|---|---|---|---|---|
| $h^C$ [m] | 160 | $\rho_i^M$ [kg m$^{-3}$] | 900 | $C_P^M$ [J kg$^{-1}$ $^\circ$C$^{-1}$] | $4.18 \times 10^3$ |
| $H^C$ [m] | 200 | $\alpha^M$ [$^\circ$C$^{-1}$] | $5.82 \times 10^{-5}$ | $L^M$ [J kg$^{-1}$] | $2.5 \times 10^5$ |
| $\overline{T_2}^C$ [$^\circ$C] | 0.8603 | $\beta^M$ [(g/kg)$^{-1}$] | $8 \times 10^{-4}$ | $\sigma^M$ [g/kg] | 30 |
| $\overline{S_2}^C$ [g/kg] | 34.7549 | $\rho_0^M$ [kg m$^{-3}$] | 1000 | $T_f^M$ [$^\circ$C] | -1.86 |
| $K^t$ [m s$^{-1}$] | $10^{-4}$ | $S_0^M$ [g/kg] | 35 | | |

## 3 Results

In Section 3.1 the forcing conditions of the MRP box model, as determined from the CESM simulation are discussed. Section 3.2 presents an analysis of the general model behaviour, with cases MKL and MKH being discussed in Section 3.3. The results for the cases PFB, PFH, and PFS are shown in Section 3.4. In the last Section 3.5, the effects of additive noise in the freshwater flux on the model behaviour for the cases MKL and PFB are described.

### 3.1 Forcing Conditions

The MRP box model is forced by a monthly varying heat flux ($Q_{oa}$ or $Q_{ia}$) and a monthly varying freshwater flux ($F$) that are repeated for each model year (Fig. 3c,d, Table 3). The monthly-averaged heat fluxes are retained from the CESM. The fluxes are spatially averaged over the polynya region (2$^\circ$E – 11$^\circ$E $\times$ 63.5$^\circ$S – 66.5$^\circ$S, black outlined region in Fig. 2) defined in van Westen and Dijkstra (2020a). The surface heat flux strongly increases when the sea-ice fraction is lower than 0.5 in the CESM (not shown). Therefore, sea-ice fractions lower (higher) than 0.5 represent a sea-ice free (covered) regime with the ocean-atmosphere (sea ice-atmosphere) heat flux noted by $Q_{oa}$ ($Q_{ia}$). Note that the MRP box model has a discrete sea-ice fraction of either 0 or 1. The monthly averaged heat fluxes are linearly interpolated in time. We do not use the sea ice-ocean flux from CESM, since this flux is already represented in the model equations (e.g. by the term $K(T - T_f)$ in equation (4)).

There is a tendency of more evaporation (decrease in $F$) when an MRP forms. Therefore, the model uses different values for the freshwater fluxes during a polynya period ($F_P$) and during a non-polynya period ($F_{NP}$); these freshwater fluxes are also retained from the CESM simulation, and, just as the heat fluxes, spatially averaged over the polynya region. The monthly-averaged freshwater fluxes are also linearly interpolated in time. The yearly freshwater input is 0.38 m and 0.24 m for non-polynya years and polynya years, respectively. The yearly freshwater input of $F = 0.38$ m is within the range presented in Martinson et al. (1981) (0.38 – 1.73 m), this range based on limited and outdated observations. These observations do not include polynya years, and therefore they are not necessarily representative for the freshwater input during a polynya event.

Values for the four horizontal advective fluxes ($F_{T1}, F_{S1}, F_{T2}$ and $F_{S2}$), background temperatures ($T_{b1}$ and $T_{b2}$) and salinities ($S_{b1}$ and $S_{b2}$) are obtained from the CESM simulation. The first layer uses a constant background temperature

**Table 3.** Ocean-atmosphere heat flux ($Q_{oa}$) in Wm$^{-2}$, sea ice-atmosphere heat flux ($Q_{ia}$) in Wm$^{-2}$, and the freshwater input (*F=P-E*) in mm/day for polynya (P) and non-polynya (NP) regimes per month determined from the CESM simulation of van Westen and Dijkstra (2020a). Positive values represent fluxes going into the ocean or the sea ice (warming and net precipitation). Negative values represent fluxes going to the atmosphere (cooling and net evaporation). All fluxes retained from the CESM simulation are spatially averaged over $2°E - 11°E$ $\times\ 66.5°S - 63.5°S$.

| Month | $Q_{oa}$[Wm$^{-2}$] | $Q_{ia}$[Wm$^{-2}$] | $F_P$[mm/day] | $F_{NP}$[mm/day] |
|-------|------|------|------|------|
| Jan | 61.4 | 61.4 | 0.91 | 0.87 |
| Feb | -23.6 | -23.6 | 1.17 | 1.06 |
| Mar | -90.8 | -90.8 | 0.96 | 1.04 |
| Apr | -144.1 | -86.1 | 0.62 | 0.86 |
| May | -161.3 | -90.3 | 0.96 | 0.93 |
| Jun | -202.3 | -79.3 | 0.47 | 1.17 |
| Jul | -246.9 | -72.5 | 0.22 | 1.09 |
| Aug | -205.6 | -65.2 | 0.13 | 1.07 |
| Sep | -76.9 | -40.3 | 0.62 | 1.27 |
| Oct | -43.0 | -1.2 | 0.69 | 1.32 |
| Nov | 107.4 | 44.1 | 0.78 | 1.11 |
| Dec | 128.2 | 128.2 | 0.44 | 0.79 |

($T_{b1} = -0.33°C$). The constant background salinity ($S_{b1}$) for the surface layer is slightly changed to tune the model (from 34.5 to 34.4818 g/kg). This is necessary to be able to simulate multiple polynya events. In the CESM, the background temperature and salinity of the subsurface layer (200 – 1000 m depths) are periodically varying as shown in Fig. 4. The dominant period in CESM is 25 years (not shown). This translates into a prescribed 25-year cycle for $F_{T2}$ and $F_{S2}$. The time mean of $T_{b2}$ and $S_{b2}$ are also shown in Fig. 4. The effect of stratified Taylor columns are assimilated in the subsurface temperature and salinity time series.

These horizontal fluxes are used to make sure the water masses in the box do not drift away from the surrounding water masses. We have used fitted background states since we are using a highly idealised model incapable of reproducing the CESM simulation accurately. Using a fitted background makes it possible to test high level hypotheses with this model. All the horizontal fluxes are dependent on a relaxation timescale ($\tau$), which is based on the advective time scale of the Weddell Gyre ($\tau_A = L/U$). The typical velocity scale in the Weddell Gyre is on the order of $5 \times 10^{-2}$ m s$^{-1}$ (Klatt et al., 2005), and the typical length scale of the Weddell Gyre is $10^6$ m. This results in an advective time scale of 230 days. To be able to represent multiple MPR events in the model, $\tau$ is chosen as (tuned to) $\frac{1}{200\ days}$.

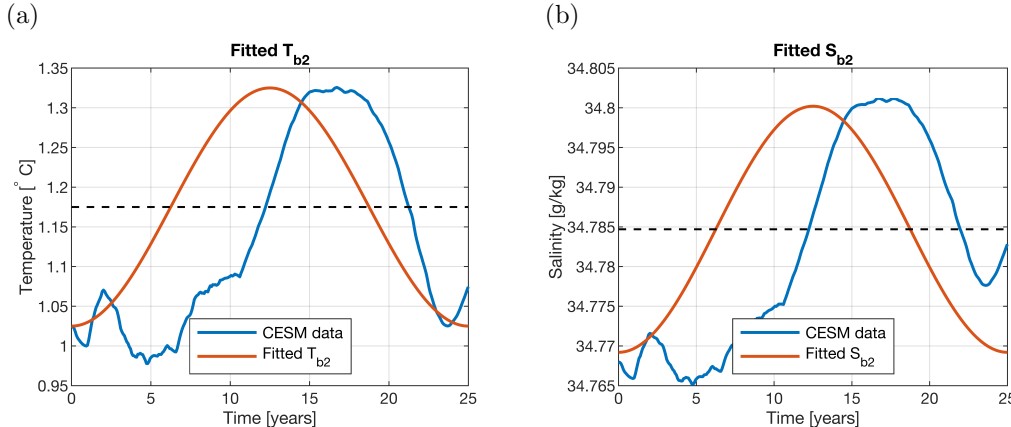

**Figure 4.** (a) Subsurface background temperature ($T_{b2}$) (red) used in the extended model set up fitted to model years 206-231 of the CESM simulation of van Westen and Dijkstra (2020a) (blue). The CESM simulation model output (blue line) is averaged over depth (200 – 1000 m), and smoothed through a 5-year running mean. The black dashed line represents the time mean used in MKL and MKH. (b) Same as (a) but now for the subsurface background salinity ($S_{b2}$). The CESM data is spatially averaged over the region of $11°E – 12°E \times 63.5°S – 66.5°S$, which is upstream of the polynya region defined in van Westen and Dijkstra (2020a)

## 3.2 Yearly Repeated Cycles

The MRP box model displays three types of yearly cycles, which are shown schematically in Fig. 5 for the surface box temperature and salinity. Note that the salinity axis has no values. If actual values would have been used, the cycles would overlap (see e.g. in Fig. 7c). In Fig. 5, each cycle starts at 'A' and follows in alphabetical order where each letter stands for a
5 regime transition. We use the following definition for MRP formation in this box model: An MRP has formed when the sea ice has melted away completely while there is still ocean cooling (negative heat flux).

For the 0-overturn cycle, the model cycles between regime II (stable, sea-ice free) and IV (stable, sea-ice covered). At 'A', the model transits from regime II → IV because the surface layer reaches the freezing temperature. During sea-ice growth, brine is rejected resulting in an increase of the surface salinity. During austral spring, the sea ice melts leading to an increase of the freshwater flux and consequently salinity levels decrease. At 'B', the model transits back to regime II since all the sea
10 ice has melted. Next, the surface layer temperature $T_1$ is mainly controlled by the atmospheric heat flux, which is positive (negative) during austral summer (autumn). When the model is cooled down to freezing temperatures, the model transits in 'A' to regime IV and the (seasonal) cycle continues.

For the 1-overturn cycle, where the model overturns once, the surface layer also reaches freezing temperatures in 'A' and
15 brine is rejected. The difference here is that the water column becomes statically unstable ($\rho_1 \geq \rho_2$) in 'B' and enters the overturning regime III (mixed, sea-ice covered) for a relatively short period (in the order of minutes) after which in transits back to regime IV in 'C'. Due to vertical mixing, relatively warm and saline subsurface water is mixed upwards and melts the sea ice; a polynya forms. During polynya formation, the model transits to regime II in 'D'. Similar to the 0-overturn cycle, the

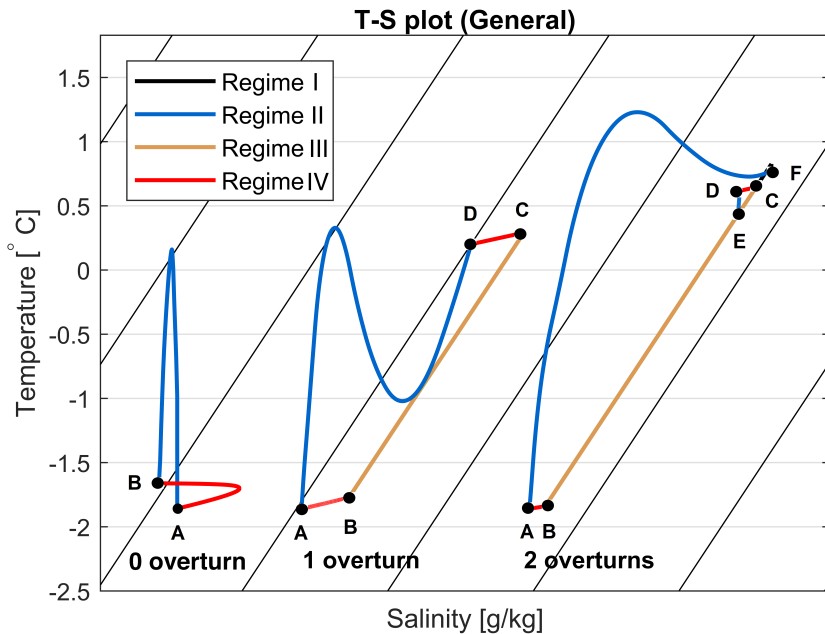

**Figure 5.** Schematic $T_1$-$S_1$ diagram with arbitrary scale on the salinity axis increasing to the right, showing three general cycles: 0-overturns (left), 1-overturn (middle), and 2-overturns (right) each year. The different model regimes are displayed in black (I), blue (II), orange (III), and red (IV). The letters (A-F) represent regime changes. The cycle starts at A, and follows the alphabetical order. The black contours represent isopycnals, and density increases from left to right. No values are used on the salinity axis for clarity as otherwise the cycles would overlap.

surface layer is controlled by the atmospheric heat flux for the remaining part of the year (austral winter – autumn). This cycle is similar to the one reported in Martinson et al. (1981).

For the 2-overturn cycle, the model transits through all four regimes. The first part of the cycle (A – D) is similar to the 1-overturn cycle and a polynya forms in 'D' but considerably less sea ice forms between 'A' and 'B' compared to the 1-overturn 5 cycle. After 'D', the surface layer is cooled during austral winter, but the difference here is that the surface layer becomes static unstable in 'E' and switches to regime I (mixed, sea-ice free). After the model state becomes statically stable in 'F', it stays stable for the remaining part of the year.

### 3.3 Cases MKL and MKH

When the MRP box model was configured with parameter values and numerical schemes as reported in Martinson et al. (1981), 10 their results could not be reproduced. Unfortunately, there is an incomplete parameter documentation in Martinson et al. (1981). In addition, it is not clear how the atmospheric heat fluxes were interpolated in time. For the cases MKL and MKH (for which

parameters are slightly different than the ones in Martinson et al. (1981)), the MRP box model is spun up for 75 years and continued for another 25 years (model years 76 – 100).

The results for the cases MKL and MKH are shown in Fig. 6a-c and Fig. 6d-f, respectively. For both cases, the density of both layers, the sea-ice thickness and a T-S diagram are shown for the last 25 years. MKL remains in the 0-overturn cycle (Fig. 5) while in MKH (larger $K_S$), a 2-overturn cycle is found. In Fig. 6b and e, the sea-ice thickness shows the same (yearly) cycle every year. For temperature and salinity in the surface layer this can be seen in Fig. 6c and f indicating that there are no variations with a period larger than 1 year. In case MKH, there is a little sea-ice growth each year followed by overturning and subsequent sea-ice melt. Using our definition of polynya formation, we find that a polynya forms each year, and therefore case MKH can be considered as having one long polynya period. In summary, for both cases MKL and MKH only yearly cycle solutions are found under the given forcing. Both solutions do not correspond with MRP events in observations, since no persistent MRP is found over a few years (Campbell et al., 2019).

In the CESM results of van Westen and Dijkstra (2020a), the MRP reappeared every 25 years and an MRP event lasted for about 6 consecutive years. Between model years 210 – 230, prior to MRP formation, no deep convection occurred, and the region was statically stable (van Westen and Dijkstra, 2020b). The temperature and salinity of the surface layer are seasonally varying between model years 210 – 230 (Fig. 7). The surface salinity values are increasing during the three years (model years 228 – 230) before MRP formation; these relatively higher surface salinity values did not initiate deep convection near Maud Rise in the CESM (van Westen and Dijkstra, 2020b). During model years 231 – 237 an MRP forms in the CESM. Deep convection mixes relatively warm and salty water from subsurface depths towards the surface resulting that the temperature and salinity of the surface layer deviate from the seasonal cycle. Clearly, the MRP box model results for the cases MKL and MKH, under the CESM-derived surface forcing, cannot reproduce the CESM results as in van Westen and Dijkstra (2020a).

### 3.4 Cases PFB, PFH and PFS

For the cases PFB, PFH and PFS, the MRP box model is spun up for 75 years and continued for another 25 years (model years 76 – 100). In case PFB (Fig. 8) both heat and salt subsurface flux forcings are included. Based on the fitted subsurface fluxes (Fig. 4), and Eq. (1), the effects of the background subsurface temperature and salinity on the density almost compensate each other. There is a relatively small subsurface density maximum between model years 87 – 88 (red line Fig. 8a). The cycle shown in Fig. 8 is repeated every 25 years, which means multidecadal recurring polynya events are simulated. Of the simulated 25 years, there are 7 polynya years and 18 non-polynya years. The polynyas are visible by reduced sea-ice maxima (mean of 0.21 m for polynya years versus a mean of 0.29 m for non-polynya years; Fig. 8b). In a 25-year cycle, the water column overturns when the subsurface density is approaching its minimum. We can also see that years with overturning can be separated by a year without overturning (e.g. in Fig. 8 in year 76 and 78 there is overturning, but not in year 77). The subsurface processes also influence the characteristics of the surface layer (Fig. 8c). In cases MKL and MKH the yearly cycles overlap each other but in the case PFB the yearly cycles are different as a response to the subsurface heat and salt accumulation which is also seen in the CESM simulation (Fig. 7).

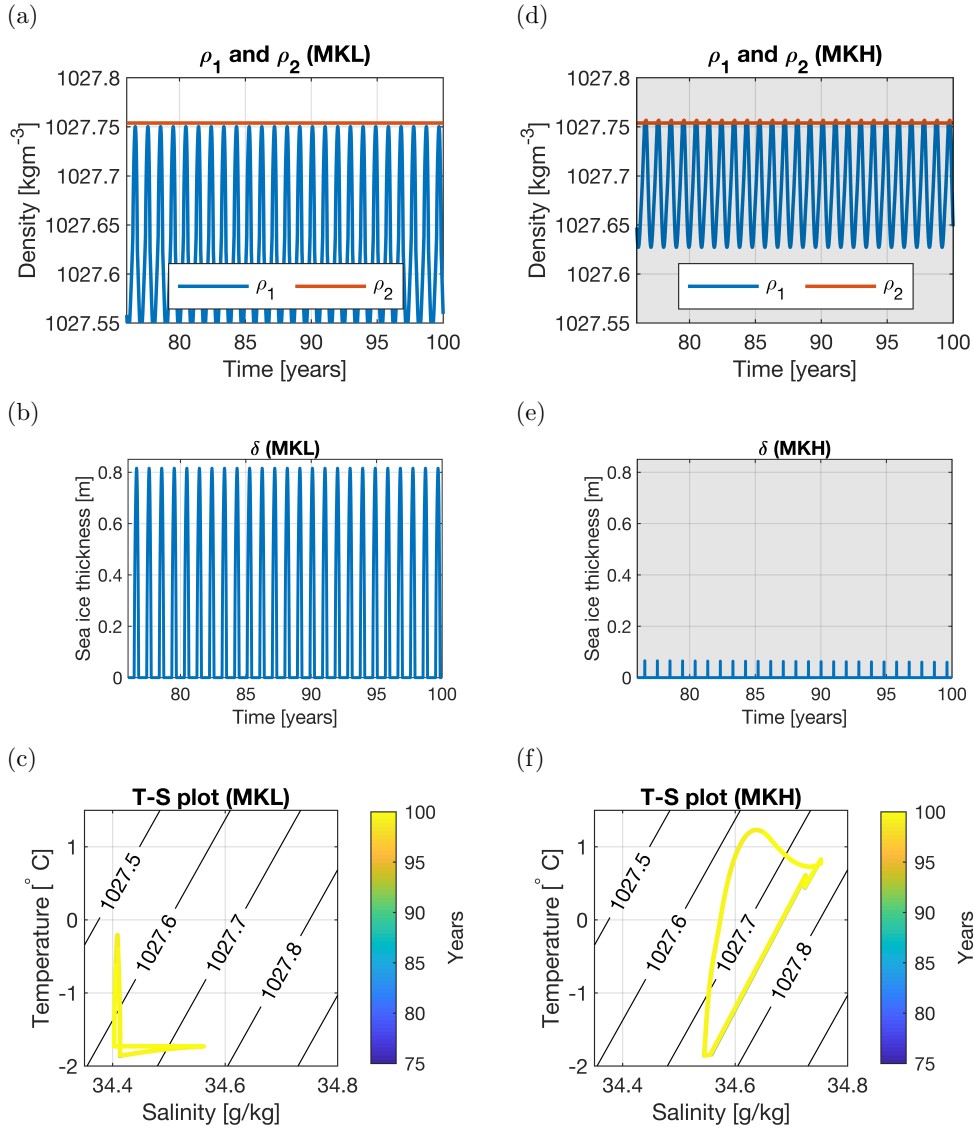

**Figure 6.** Years 76-100 for case MKL (a-c; $K_S = 1.375 \times 10^{-6}$ ms$^{-1}$) and MKH (d-f; $K_S = 2 \times 10^{-6}$ ms$^{-1}$) . (a, d) Density of the surface (blue) and subsurface (red) layer. (b, e) Sea-ice thickness. (c, f) T-S plot of the temperature and salinity of the surface layer. Colouring of the curves represents time, ranging from year 76 (blue) to year 100 (yellow). Only the last year is visible, because previous years have the same yearly cycle. The direction in time is the same as in Fig. 5. The black contour lines represent the density in kg m$^{-3}$. Shading represents polynya years.

Whereas PFB uses both subsurface fluxes (heat and salt), case PFH (Fig. 9) only uses a time-varying subsurface heat flux forcing. By thermal expansion, the subsurface density decreases and the water column becomes statically unstable in model

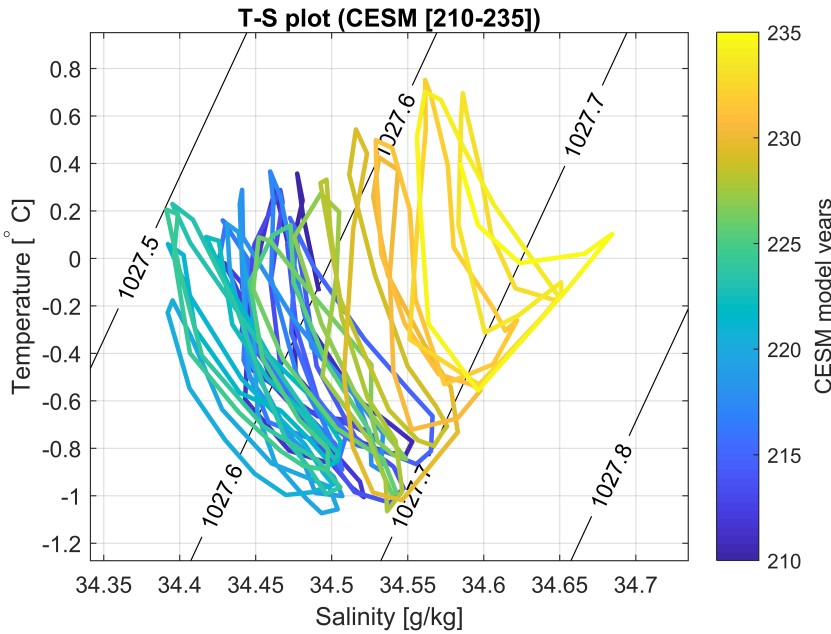

**Figure 7.** T-S plot of model years 210 – 235 from the CESM simulation (van Westen and Dijkstra, 2020a). The model is based on monthly values of the temperature and salinity averaged over the surface layer (0 – 160m) over the Polynya region (2°E – 11°E × 66.5°S - 63.5°S). The colour coding represents time (year 210 is blue, year 235 is yellow). The polynya period captured in this plot is between years 231 (orange) and 235 (yellow). The black contour lines represent density in kg m$^{-3}$, using a simple linear equation of state (Eq. (1)).

year 86. During vertical mixing, relatively low sea-ice fractions are found compared to the stable model years. Of the simulated 25 years, there are 13 non-polynya years and 12 polynya years. Note that here we do have one long polynya period, which is different from what we see in the PFB case (Fig. 8). In case PFS (Fig. 10), only a time-varying salt subsurface flux forcing is considered. By haline contraction, the subsurface density increases and the water column is statically stable during relatively

5   high levels of subsurface salinity. When subsurface salinity levels decrease over time, the water column becomes unstable and a polynya forms during model years 76 – 84 and years 94 – 100. Of the simulated 25 years, there are 10 non-polynya years and 15 polynya years.

In all three cases with subsurface forcing, the MRP box model is able to simulate the general features also seen in the CESM simulation (van Westen and Dijkstra, 2020a). These cases show a repeating 25-year cycle, which is the same period

10  as the period of the subsurface forcing and the same period as seen in CESM. Where CESM has more non-polynya years than polynya years case PFS have more polynya years, and case PFH has as many non-polynya years as polynya years. Case PFB, however, approaches the ratio non-polynya years versus polynya years seen in the CESM. Besides this difference, also the timing of the first overturn after a non-polynya period is different with respect to CESM. In CESM the first overturn occurs approximately 6 years after the subsurface heat and salt accumulation have reached their maximum. PFS overturns

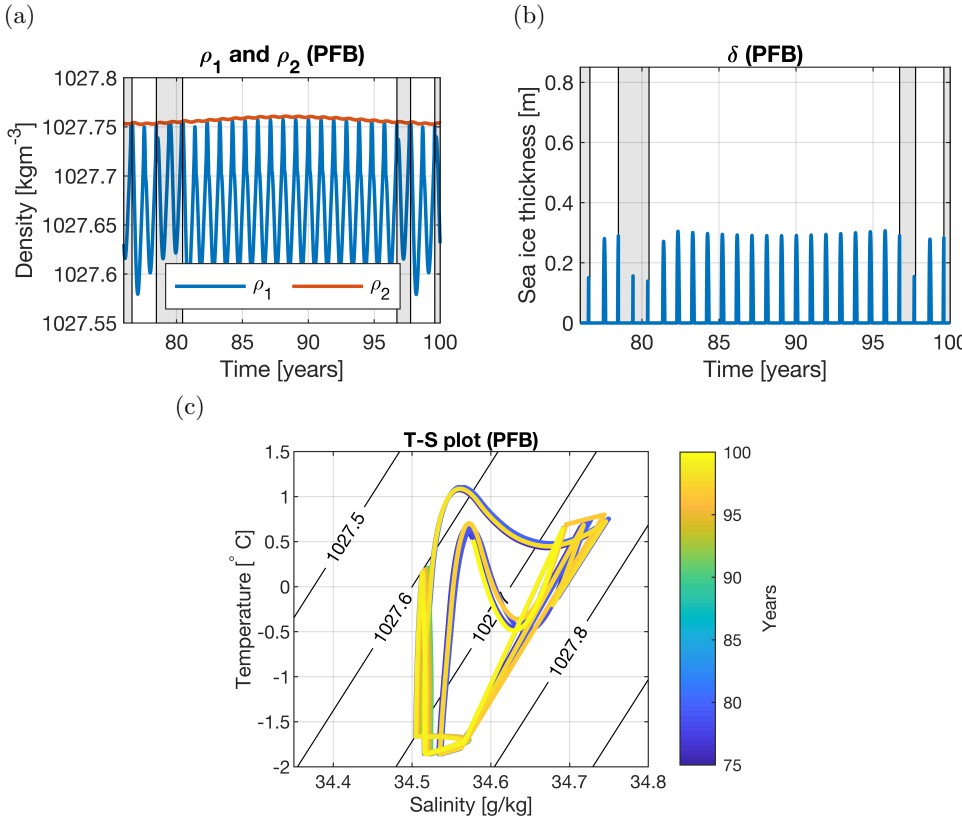

**Figure 8.** Years 76-100 for case PFB (both subsurface fluxes). (a) Density of the surface (blue) and subsurface (red) layer. Shading represents polynya years. (b) Sea-ice thickness. Shading represents polynya years. (c) T-S plot of the temperature and salinity of the surface layer. Colouring of the lines represents time, ranging from year 76 (blue) to year 100 (yellow). The black contour lines represent the density in kg m$^{-3}$. Polynyas are present between year 76 (blue), years 78-81 (blue to cyan), years 96-97 (orange), and year 99 (yellow).

in approximately the same year, while PFB overturns 3 years later. Case PFH differs most, since it overturns 8 years earlier, and even before the subsurface heat accumulation has reached its maximum. These differences are probably caused by the idealizations in the MRP box model, and most likely due to the representation of mixing in this model, compared to that in CESM.

## 3.5  Atmospheric Variability

In this section we analyse whether the multidecadal MRP variability, as found for the cases PFB, PFH and PFS, is robust under the influence of atmospheric variability, such as intense winter storms. This atmospheric variability is incorporated into the MRP box model by adding white noise to the surface freshwater flux. White noise was added as in Eq. (6):

$$F_N(t) = F(t) + \sigma_N r(t) \tag{6}$$

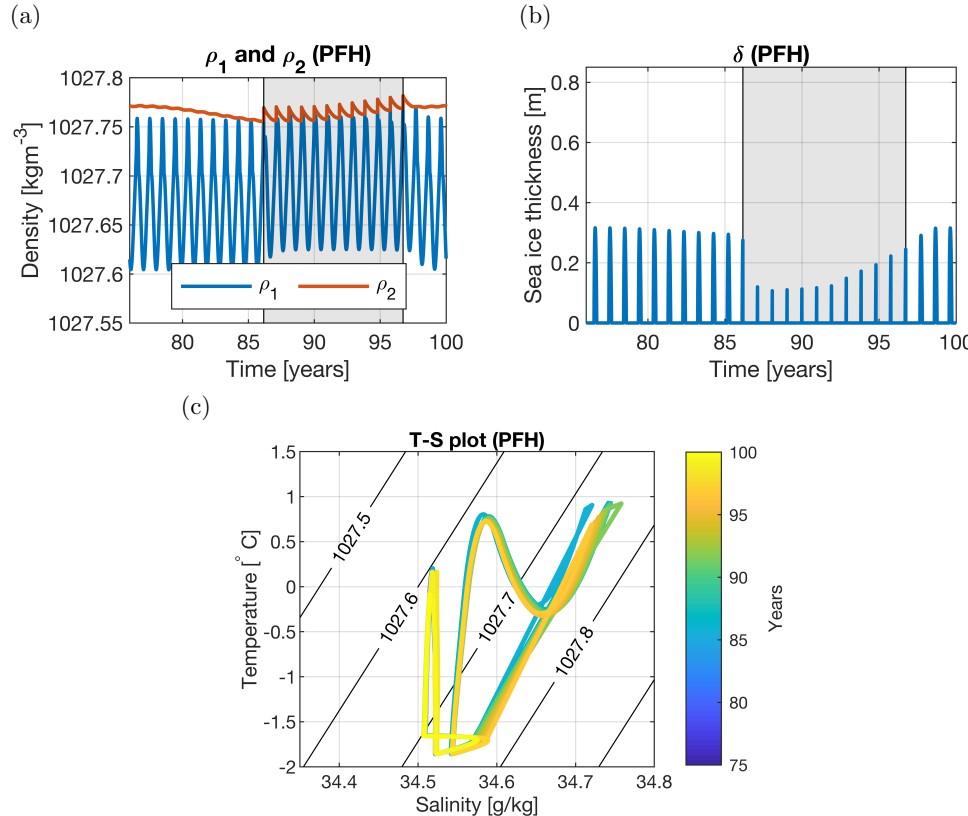

**Figure 9.** Years 76-100 for case PFH (with only a subsurface heat flux). (a) Density of the surface (blue) and subsurface (red) layer. Shading represents polynya years. (b) Sea-ice thickness. Shading represents polynya years. (c) T-S plot of the temperature and salinity of the surface layer. Colouring of the lines represents time, ranging from year 76 (blue) to year 100 (yellow). The black contour lines represent the density in kg m$^{-3}$. Polynyas are present between year 86 (cyan) and year 96 (orange).

where $F_N$ is the freshwater flux with noise, $F$ is the freshwater input without noise as in Fig. 3d, $\sigma_N$ the standard deviation of the noise (0.6613 mm/day), and $r$ is a random draw from a standard normal distribution on every time step. The standard deviation of the noise was determined from the CESM simulation.

Figure 11 displays the spectral power of the variables $T_1$ (Fig. 11a), $S_1$ (Fig. 11b), and $\delta$ (Fig. 11c) for case PFB. For all variables, the percentile, mean and median, the dominant period is about 25 years, the same period as the subsurface forcing and is clearly visible in all variables (Fig. 11a). The single ensemble member also shows a dominant period of approximately 10 years, showing that the noise can also induce shorter periods of convection. Using the same white noise forcing in case MKL yields no dominant multidecadal period (not shown). As was seen in Section 3.3, the MKL case remains in 0-overturn cycle without polynyas (Fig. 6). Atmospheric noise can cause polynyas in the MKL case. However, when MKL is forced into a polynya state, it cannot be forced out of the polynya state; a polynya forms every year. This means MKL with noise will in

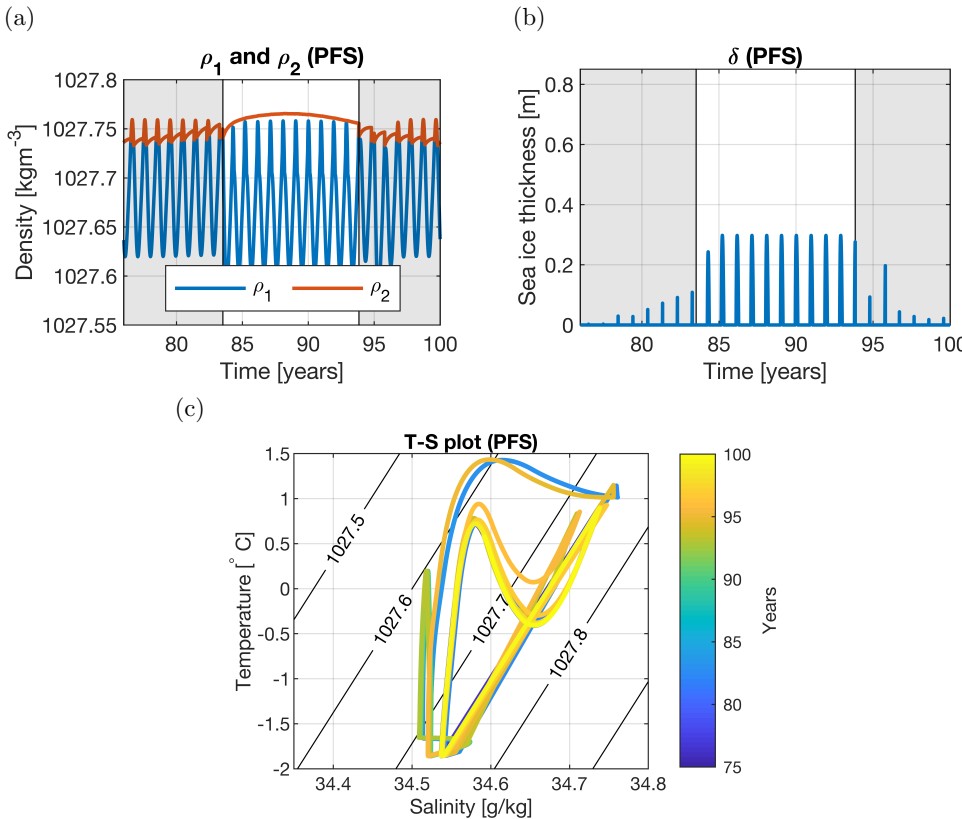

**Figure 10.** Years 76-100 for case PFS (with only a subsurface salt flux). (a) Density of the surface (blue) and subsurface (red) layer. Shading represents polynya years. (b) Sea-ice thickness. Shading represents polynya years. (c) T-S plot of the temperature and salinity of the surface layer. Colouring of the lines represents time, ranging from year 76 (blue) to year 100 (yellow). The black contour lines represent the density in kg m$^{-3}$. Polynyas are present between year 76 (blue) and 83 (cyan), and after year 94 (orange to yellow).

time show the same behaviour as MKH without noise. The combination of PFB and MKL shows that atmospheric variability can alter the timing of polynya formation but the dominant period is set by subsurface fluxes of heat and salt.

## 4    Summary and discussion

In this study, an extended version of the idealized box model of Martinson et al. (1981), was used to investigate the importance of surface forcing and subsurface forcing on Maud Rise Polynya (MRP) formation, with the aim to understand the CESM results in van Westen and Dijkstra (2020a) in more detail. The extensions in our MRP box model with respect to Martinson et al. (1981) are a dynamic subsurface layer, and horizontal subsurface heat and salt fluxes to both the surface and the subsurface layer. Even though the results in Martinson et al. (1981) could not be reproduced exactly (due to incomplete information in the

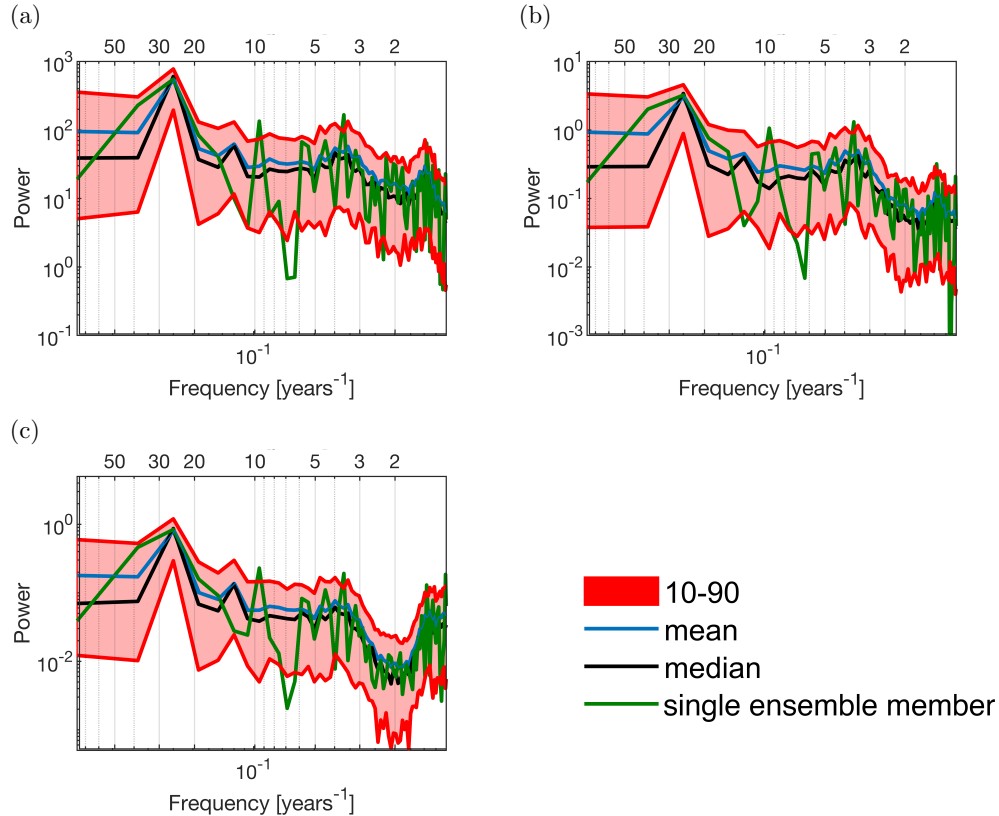

**Figure 11.** Spectral analysis for variables (a) $T_1$, (b) $S_1$, and (c) $\delta$ for case PFB (both subsurface fluxes). The analysis is based on 100 ensemble members. Each ensemble member contains the last 75 years of a 100 year run to exclude spin up effects. The red band represents the ensemble members between the $10^{th}$ and $90^{th}$ percentile. Also the mean (blue), median (black) and a single ensemble member (green) are displayed. Both axes are on log scale. The top x-axis displays the period in years, the bottom x-axis the frequency in years$^{-1}$.

original paper), the qualitative behaviour of the MRP box model (reduced to the case used in Martinson et al. (1981)) was the same.

The results for the cases MKL and MKH (close to the case in Martinson et al. (1981)) show that deep convection is caused by brine rejection in a preconditioned surface layer. Brine rejection causes a rapid increase of density in the surface layer. The results (Fig. 6d-f) clearly show that this eventually induces deep convection. However, brine rejection alone cannot explain observed multiple polynya events (e.g. the 1970s and 2017 events), since brine rejection is present in all years with sea-ice growth, and not all years show deep convection and subsequent polynya formation (Fig. 6a-c). In MKL, there is no overturn (Fig. 6a-c) and in case MKH, two overturns occur each year (Fig. 6d-f). In Martinson et al. (1981), it was also shown that in time the model state will reach a yearly repeating cycle, either in regimes II (sea-ice free, stable) and IV (sea-ice covered, stable) (0-overturn cycle), or in regimes I (sea-ice free, mixed) and II (sea-ice free, stable) (2-overturn cycle). The 0-overturning case can be explained by a too strong stratification. In case MKL, the salt transfer (governed by $K_S$) of the subsurface layer to

the surface layer is too weak to overcome the stratification. In the second solution (two overturns each year) this salt transfer towards the surface is increased, which causes static instability in the water column with mixing as a result.

In climate models, such as CESM, the water column stabilises after deep convection when the heat and salt reservoirs are depleted. This depletion leads to stabilisation of the water column by increasing the density of the subsurface layer through heat depletion. This physical process is missing in cases MKL and MKH, and therefore the MRP box model state is not able return to a non-polynya regime. However, in Martinson et al. (1981), a solution was shown with overturns in the first years, after which a stable, non-polynya cycle appeared. These overturns in the first years are the result of the initial conditions. When the model is forced by the CESM surface forcing, either a polynya forms each year (MKH) or the model stays in a stable non-polynya state (MKL). Clearly, the cases MKL and MKH cannot capture the behaviour of the CESM as found in van Westen and Dijkstra (2020a).

In cases PFB (Fig. 8), PFH (Fig. 9) and PFS (Fig. 10), subsurface forcing derived from CESM was prescribed in the MRP box model. The results showed periodic polynya events with the same dominant period as seen in van Westen and Dijkstra (2020a) caused by the periodic subsurface forcing. The subsurface forcing preconditions both the subsurface and the surface layer, after which brine rejection is essential to induce deep convection. Note that in the CESM results of van Westen and Dijkstra (2020b), brine rejection was less important as convection was initiated at the subsurface. Because this subsurface-iniated convection in CESM is spatially localized, we probably do not find it in the MRP box model. The finding of van Westen and Dijkstra (2020b) that the subsurface heat flux is more dominant than the subsurface salt flux is not confirmed in the box model, since all cases PFB, PFH, and PFS show comparable behaviour. The subsurface heat flux, however, is expected to be dominant as it influences every quantity ($T_2$, $\rho_2$, $T_1$, $\rho_1$, $\delta$, $S_1$, $S_2$) in the model. The subsurface salt flux only affects the density and salinity of both layers, and therefore is expected to have a much smaller influence on the results. Kaufman et al. (2020) also found heat build up in the ocean and attributed this to reduced heat loss under sea-ice covered conditions. Ocean heat advection actually seemed to counteract the heat build up. In our model, such a situation does not occur, since $T_2$ is always larger than $T_{b2}$ because the subsurface layer is losing heat to the surface layer.

The MRP box model is able to capture the general features of MRP formation as seen in van Westen and Dijkstra (2020a) and shows the importance of the subsurface forcing. However, the model is still too idealised to accurately capture the precise MRP formation processes in the CESM simulation. The asymmetry in the non-polynya regime versus the polynya regime was poorly captured is cases PFH and PFS. The asymmetry in case PFB compared best to the CESM but still showed relatively more polynya years compared to the CESM simulation. This is probably due to the difference in how vertical mixing is represented. In the MRP box model the layers are either in a stably stratified configuration with a constant layer depth, or they are completely mixed. In van Westen and Dijkstra (2020a), a KPP boundary mixed layer scheme is used. Representing the growth of the mixed layer more accurately would improve the model, and possibly would lead to a better representation of the asymmetry between the two regimes. When the mixed layer is allowed to grow more gradually, a lag is introduced in the system. This will delay the formation of a polynya. Due to this instant mixing, both temperature and salinity in the surface layer increase instantly. This results in large differences after overturning between the MRP box model results and the CESM simulation results (van Westen and Dijkstra, 2020a).

Martinson et al. (1981) were the first to investigate the processes responsible for polynya formation. Using their box model they suggested that surface processes are responsible for polynya formation, a view that is still widely supported nowadays. What we have shown is that their adjusted model to the Maud Rise region is not capable of simulating multiple polynya events as seen in observations. When the model is extended with variable subsurface heat and salt accumulation, the model is capable of simulating multiple MRP events and able to qualitatively reproduce the CESM results of van Westen and Dijkstra (2020a). Our study suggests that surface related processes cannot completely explain MRP formation nor long-term MRP variability, and that subsurface advective processes need to be taken into account.

*Code and data availability.* The model code and input files of the conceptual box model are available via GitHub: https://github.com/dboot0016/MRP_Conceptual_box_model. CESM model data are available upon request from the corresponding author.

## Appendix A: Regime transitions

In this appendix the conditions of the regime transitions are shown. A regime transition changes the initial conditions for the new regime. The new initial conditions are indicated with a prime. Horizontal bars above a variable represent averaging over the water column due to overturning: $\bar{X} = (hX_1 + (H-h)X_2)/H$, where $X$ is either $T$ or $S$.

$$\text{regime I} \rightarrow \text{regime II when } -\alpha\frac{dT}{dt} + \beta\frac{dS}{dt} < 0;$$
$$T_1' = T, S_1' = S, \delta' = 0, T_2' = T, S_2' = S;$$
$$\text{regime I} \rightarrow \text{regime III when } T = T_f;$$
$$T' = T_f, S' = S, \delta' = 0;$$
$$\text{regime II} \rightarrow \text{regime I when } \rho_1 = \rho_2;$$
$$T' = \bar{T}, S' = \bar{S}, \delta' = 0;$$
$$\text{regime II} \rightarrow \text{regime IV when } T_1 = T_f;$$
$$T_1' = T_1, S_1' = S_1, \delta' = 0, T_2' = T_2, S_2' = S_2;$$
$$\text{regime III} \rightarrow \text{regime I when } \delta = 0;$$
$$T' = T, S' = S, \delta' = 0;$$
$$\text{regime III} \rightarrow \text{regime IV when } -\alpha\frac{dT}{dt} + \beta\frac{dS}{dt} < 0;$$
$$T_1' = T, S_1' = S, \delta' = \delta, T_2' = T, S_2' = S;$$
$$\text{regime IV} \rightarrow \text{regime II when } \delta = 0;$$
$$T_1' = T_1, S_1' = S_1, \delta' = 0, T_2' = T_2, S_2' = S_2;$$
$$\text{regime IV} \rightarrow \text{regime III when } \rho_1 = \rho_2;$$
$$T' = \bar{T}, S' = \bar{S}, \delta' = \delta;$$

*Author contributions.* D.B. developed the model code, performed the numerical experiments, and analyzed the data. D.B. wrote the manuscript with input from all authors. R.M.W. and H.A.D. conceived the idea of the study and were in charge of overall direction and planning.

*Competing interests.* The authors declare that they have no conflict of interest.

*Acknowledgements.* The authors thank Michael Kliphuis (IMAU, UU) for performing the CESM simulations. The computations were done
5 on the Cartesius at SURFsara in Amsterdam. Use of the Cartesius computing facilities was sponsored by the Netherlands Organization for Scientific Research (NWO) under the project 15556.

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
