# Peer review of "Multidecadal Polynya Formation in a Conceptual (Box) Model"

_Ocean Science, 2020_

## Referee Comment (RC1) · David Bailey (Referee) · 11 Aug 2020

This is an interesting study where the authors have developed a box model formulation similar to Martinson et al. 1981 for upper ocean mixing in the region of Maud Rise. While this is a very nice study, I am concerned about the underlying assumption that is basing polynya formation on CESM high resolution model results. Here are some specific concerns that I have with the manuscript as is.

1. First off, it is extremely difficult to figure out exactly how these high resolution simulations were done. The authors refer to a manuscript in review (van Westen et al. 2020) for a description of the experiments. However, the model experiments and configuration are actually described in an earlier manuscript by van Westen and Dijkstra 2017.

Please add more detail here about these simulations so the reader does not have to sift through the rest of the literature. Can the authors also comment about using year 2000 forcing as a control run for 250 years which is not a balanced climate?

2. Here is my biggest concern. Based on the high resolution CESM simulations that I have seen (McClean et al. 2011; Kirtman et al. 2012; Small et al. 2014; Chang et al. (2020)) the mean state of the Antarctic sea ice is biased thin and not extensive enough. This I believe is one of the main reasons the polynyas do not show up in low resolution simulations, but do in the high resolution. That is, I believe that the polynyas are a result of a mean state bias. I realize this is more relevant to the van Westen et al. 2020 manuscript, but I think this should be addressed here as well. Also, this is a bit of semantic issue. Most of the "polynyas" that form in these high resolution simulations are sort of closed off embayments. As the ice grows in the SH, the Weddell gyre circulates sea ice to the East and eventually it meets up with the Maud Rise coastal area and encloses an open-ocean region. A polynya in my mind is when the area is completely sea ice covered in mid-winter and a hole opens up in the sea ice. Look at animations of daily sea ice concentration. The seasonality is the key here. At least stating the assumption that while polynya formation and the frequency of actual polynya events versus embayments in the CESM in these simulations may not be realistic, these are the processes behind this particular model simulation.

Chang et al. (2020) Under review iHESP project paper.

Kirtman, B. P., et al. (2012), Impact of ocean model resolution on CCSM climate simulations, Clim. Dyn., 39, 1303–1328.

McClean, J., et al. (2011), A prototype two‐decade fully‐coupled fine‐resolution CCSM simulation, Ocean Model., 39, 10–30.

Small, R. J., et al. (2014), A new synoptic scale resolving global climate simulation using the Community Earth System Model, J. Adv. Model. Earth Syst., 6, 1065–1094, doi:10.1002/2014MS000363.

3. The box model description is very confusing. Some of the terms in the equations are not described. While this might be in the Martinson et al. 1981 paper, some more detail should be repeated here. I guess the Martinson paper came up with the convention of h and H-h for the layer thicknesses. I would prefer h1 and h2 here. Similarly Regime I and III only have T, S, and rho instead of T1, S1, rho1, and T2, S2, and rho2. The Tb1, Sb1, and Tb2, Sb2 variables are introduced in the equations but not explained. I see these are mentioned later on in section 3.1. I think it is also very important to highlight what is different in the model description section from the Martinson et al. 1981 paper. Is it just that you used basically the same model, but with different forcing?

The rest of my points are fairly minor.

4. What about Qio? I think this is more important when there is ice present in terms of forcing the ocean rather than Qia. Or does Qoa include Qio somehow? You have Qio from the CESM simulations already.

5. Are there any other freshwater flux observations around Antarctica? Are these open ocean only or ice-ocean?

6. I'm curious why you used fitted background T and S. You have the data from the CESM run, so why not use that?

7. Figure 4 is missing labels on the density contours and the salinity axis.

8. Figure 5 (and others). Why did you plot thickness as a measure of polynya presence? The definition is concentration based. Panels b and e in Figure 5 are not that helpful. The thickness is the same every year. The "shading" indicates there is a constant polynya in panel e right? Where on the T-S curve is the "active" polynya. It should only be during Regime IV, i.e. while on the "straight" line between density 1027.8 and 1027.7? Actually, how can you say MKH is a polynya? It looks ice free the whole year? I'm very confused here.

9. Figure 6. Can you indicate the actual polynya years in CESM here? Is it every year?

Similarly for Figure 7.

10. In the summary and discussion, the authors mention that this box model is "slightly extended". This needs to be expanded. You could not replicate the results from Martinson as I understand it. More detail here on what is new about your study!

11. Also, I sort of feel like it is missing a big "punchline". What have you added to the body of literature on Maud Rise polynyas here? Is it just the enhanced role of subsurface heat accumulation? The results from the van Westen work were not simulated with the box model, but I think more needs to be added here to explain what the box model gives you and adds to the story.

---

## Referee Comment (RC2) · Wilbert Weijer (Referee) · 17 Aug 2020

Review of: "Multidecadal Polynya Formation in a Conceptual (Box) Model", by Boot, van Westen, and Dijkstra.

In this paper the authors explore polynya behavior in a conceptual model. In particular, they focus on the role of heat accumulation at depth in preconditioning the area for deep convection.

I found the paper well-written, well-illustrated, and the research was carried out carefully. The research may not be a major advance of the field, but it's a nice contribution to the recent literature on polynyas. I don't have any main concerns, and would like

to congratulate the authors on this nice work. I only have minor comments, which, I'm sure the authors can address satisfactorily; and a point for discussion.

p.1, l. 12: Usually a distinction is made between MRPs, which are clearly related to bathymetry; and the larger Weddell Sea Polynyas (WSPs) which are not related to bathymetry, as exemplified by those observed in the mid-70s. I suggest that the authors note this distinction.

p.3, l. 2: vertical -> vertically stacked.

p.3, l. 27: remove 'a'.

p. 5-6: In my opinion, the model description is adequate, maybe with the exception of the sea ice equation, which could use some clarification.

p. 9, Caption Table 2: What do the 'bars' refer to? Overbars?

p. 9, l. 21: So are these fluxes averaged over the polynya region?

p. 10, l. 7: It would probably be good to explicitly state that this is a prescribed 25-yr cycle.

p. 10, l. 9: I think it would be good to have a better justification of the advective terms somewhere. A source of heat or salt is of course a consequence of a /divergence/ of advective fluxes. Maybe a better paradigm is that the lower box is 'bathing' in a water mass with ambient temperature $T\_b2$ and salinity $S\_b2$.

p.10, l. 16: I suspect you mean ocean cooling, so heat transfer from the ocean to atmosphere. This would mean warming of the atmosphere.

p. 11, Fig. 3: I'm a bit concerned by the strong variations, especially in the later years. I assume that the authors have checked that this water mass was not influenced by a polynya in the CESM? Evidently, you want to force the box model with upstream conditions.

p. 15, l. 25-29: Maybe you can leave out the inclusion of the factor 35 (or discuss it somewhere else)? As it stands, $F_N$ is /not/ a freshwater flux, as claimed in l. 27, but a salt flux. Besides, it would result in a sign error.

p. 20-21: I think we are missing some rules for $T_2$ and $S_2$ in certain transitions.

Discussion point:

In our recent paper (Kaufman et al. 2020) we studied the heat content in E3SMv0-HR (a close clone of CESM1), and also found that heat build-up preceded polynya formation. However, our analysis suggests that this heat build-up is driven by a reduced surface heat loss under ice-covered conditions, and not an enhanced ocean heat import (Fig. 8c). In fact, ocean heat advection appeared to counteract the heat accumulation by removing excess heat. I suppose that in the context of this box model, this situation would be represented by $T_2 > T_{b2}$ for long periods of time without polynyas. Does a situation like this occur in your model, and can you discuss the context of these occurrences?

Reference: Kaufman, Z.S., Feldl, N., Weijer, W. and Veneziani, M., 2020. Causal Interactions between Southern Ocean Polynyas and High-Latitude Atmosphere–Ocean Variability. Journal of Climate, 33(11), pp.4891-4905.

---

## Editor Comment (EC1) · Andrew Moore (Editor) · 22 Sep 2020

Dear Daan,

Your responses to the reviewer comments are, in my opinion, satisfactory, and your proposed revisions to the manuscript are appropriate at this stage.

I therefore encourage you to submit a revised version of your manuscript.

Regards,

Andrew Moore (Topic Editor)

---

## Author Response (AR3)

**MS-No.:** OS-2020-63

**Title:** Multidecadal Polynya Formation in a Conceptual (Box) Model

**Author(s):** Daan Boot, René M. van Westen and Henk A. Dijkstra

**Point-by-point reply to reviewer #1**

**November 2, 2020**

We thank David Bailey for his careful reading and for the useful comments on the manuscript.

Specific concerns:

1. *First off, it is extremely difficult to figure out exactly how these high resolution simulations were done. The authors refer to a manuscript in review (van Westen et al. 2020) for a description of the experiments. However, the model experiments and configuration are actually described in an earlier manuscript by van Westen and Dijkstra 2017. Please add more detail here about these simulations so the reader does not have to sift through the rest of the literature. Can the authors also comment about using year 2000 forcing as a control run for 250 years which is not a balanced climate?*

   **Author's reply:**
   In the revision we will include more information on the simulation of the Community Earth System Model (CESM). The results in van Westen and Dijkstra (2017) only cover the first 200 years of the simulation, while we use a period between model years 150 – 250. The complete CESM simulation (300 years) and full details can be found in van Westen et al. (2020) (https://doi.org/10.1038/s41598-020-71563-0).

   A present-day forcing (of the year 2000) is closer to current observations compared to pre-industrial control simulations (such as in CMIP). The author is right that the model is not in equilibrium, but the model has a spin-up period of 150 years. The upper ocean temperatures (1000 m) are fairly in equilibrium (see Figure S2 in van Westen et al. (2020)). Any drift can be removed by subtracting a linear or quadratic trend.

Note that pre-industrial control simulations of CMIP also are not in equilibrium, as the deep ocean fields take much longer time (millennial timescales) to equilibrate.

**Changes in manuscript:**
We will include a brief summary of the CESM simulation and equilibration and motivate the present-day configuration. For a complete overview of the full CESM results, we will refer to van Westen et al (2020).

2. *Here is my biggest concern. Based on the high resolution CESM simulations that I have seen (McClean et al. 2011; Kirtman et al. 2012; Small et al. 2014; Chang et al. (2020)) the mean state of the Antarctic sea ice is biased thin and not extensive enough. This I believe is one of the main reasons the polynyas do not show up in low resolution simulations, but do in the high resolution. That is, I believe that the polynyas are a result of a mean state bias. I realise this is more relevant to the van Westen et al. 2020 manuscript, but I think this should be addressed here as well. Also, this is a bit of semantic issue. Most of the polynyas that form in these high resolution simulations are sort of closed off embayments. As the ice grows in the SH, the Weddell gyre circulates sea ice to the East and eventually it meets up with the Maud Rise coastal area and encloses an open-ocean region. A polynya in my mind is when the area is completely sea ice covered in mid-winter and a hole opens up in the sea ice. Look at animations of daily sea ice concentration. The seasonality is the key here. At least stating the assumption that while polynya formation and the frequency of actual polynya events versus embayments in the CESM in these simulations may not be realistic, these are the processes behind this particular model simulation.*

   *Chang et al. (2020) Under review iHESP project paper.*
   *Kirtman, B. P., et al. (2012), Impact of ocean model resolution on CCSM climate simulations, Clim. Dyn., 39, 1303â1328.*
   *McClean, J., et al. (2011), A prototype two-decade fully-coupled fine-resolution CCSM simulation, Ocean Model., 39, 10â30.*
   *Small, R. J., et al. (2014), A new synoptic scale resolving global climate simulation using the Community Earth System Model, J. Adv. Model. Earth Syst., 6, 1065â1094, doi:10.1002/2014MS000363.*

**Author's reply:**
Thank you for the references. In van Westen and Dijkstra (2020a) they analyse a companion CESM simulation at a 1° resolution for 1300 years. One major difference between a low-resolution CESM (LR-CESM) and high-resolution CESM (HR-CESM) is the background stratification. The background stratification is much stronger in the LR-CESM, hence no deep convection developed over Maud Rise. This does not imply that no convection events occur in a low-resolution model. Dufour et al. (2017) demonstrate a different background stratification between a high- and low-resolution climate model which alters the periodicity of deep convection in the Weddell Sea.

Regarding the sea-ice thickness bias, van Westen and Dijkstra (2020b, https://os.copernicus.org/preprints/os-2020-33/) show the climatology of the sea-ice thickness over Maud Rise (their Figure 1d). The August sea-ice thickness is varying between 30 – 80 cm (90% range) with a time mean of about 53 cm for non-polynya years. Such values are also reported in an observation-model study in this part of the Weddell Sea (Holland et al. (2014), https://doi.org/10.1175/JCLI-D-13-00301.1). These daily-averaged (and also monthly-averaged) sea-ice fields clearly show that the polynya appears within the sea-ice and not by embayment as suggested by the reviewer.

**Changes in manuscript:**
We will discuss the results of the references provided by the reviewer regarding the potential biases. We will include an analysis of the daily-averaged sea-ice fields for model year 231 to show how these polynyas form in the CESM.

3. *The box model description is very confusing. Some of the terms in the equations are not described. While this might be in the Martinson et al. 1981 paper, some more detail should be repeated here. I guess the Martinson paper came up with the convention of h and H-h for the layer thicknesses. I would prefer h1 and h2 here. Similarly Regime I and III only have T, S, and rho instead of T1, S1, rho1, and T2, S2, and rho2. The Tb1, Sb1, and Tb2, Sb2 variables are introduced in the equations but not explained. I see these are mentioned later on in section 3.1. I think it is also very important to highlight what is different in the model description section from the Martinson et al. 1981 paper. Is it just that*

*you used basically the same model, but with different forcing?*

**Author's reply:**
The discussion of the equations was indeed quite minimal. The different terms in the equations will be discussed more elaborately. We will pay special attention to the sea-ice equation (following reviewer comment 2), and the horizontal advective fluxes related to $T_{b1}, T_{b2}, S_{b1}$ and $S_{b2}$.

The convention of $h$, and $H - h$ is indeed from Martinson et al. (1981). The suggestion to change this to $h_1$ and $h_2$ for the two layers is followed with a depth $H$ of the total layer.

The extensions to the Martinson model are: a dynamic subsurface layer, the horizontal advective fluxes, the forcing, and some parameter values have been changed. This issue is addressed in section 3.1. We will include this also in section 2.1.

**Changes in manuscript:**
The equations will be discussed more elaborately. The convention for the layer depths will be changed. The extensions/changes made with respect to the Martinson model will (also) be addressed in section 2.1.

Minor comments:

1. *What about Qio? I think this is more important when there is ice present in terms of forcing the ocean rather than Qia. Or does Qoa include Qio somehow? You have Qio from the CESM simulations already.*

   **Author's reply:**
   $Q_{io}$ is modelled via a heat transfer flux given by the term: $\rho_0 \times C_p \times K(T - T_f)$ in equations 4a, 4c, 5a and 5c (as was done in Martinson et al., 1981). This means $Q_{oa}$ does not include $Q_{io}$.

   **Changes in manuscript:**
   In the model description (2.1) the equations will be discussed more elaborately. There it will also be made clear that this term represents the heat flux between the sea ice and the ocean.

2. *Are there any other freshwater flux observations around Antarctica? Are these open ocean only or ice-ocean?*

   **Author's reply:**
   There are several observations, see for example in Trenberth et al. (2007) where they show evaporation minus precipitation in their Figure 3. The value seen there, corresponds to the value used in this paper. This is based on ERA-40 data. This includes reanalyzed data for open ocean and ice-ocean.

   **Changes in manuscript:**
   No changes necessary.

3. *I'm curious why you used fitted background T and S. You have the data from the CESM run, so why not use that?*

   **Author's reply:**
   We are using a highly idealized model and it is not suitable to reproduce the CESM simulation accurately. We believe the model is suitable to test high level hypotheses: a subsurface accumulation of heat is important for polynya formation, and a periodic subsurface accumulation of heat results in periodic polynya formation. It is more suitable to use an idealized subsurface heat and salt flux for the model than the noisy CESM data.

   **Changes in manuscript:**
   The above reasoning will be included in the revised text.

4. *Figure 4 is missing labels on the density contours and the salinity axis.*

   **Author's reply:**
   This was done purposefully to clearly show the different cycles in the T-S space. If we would have used the actual values for salinity and density, the plots would overlap, making the plot unclear (see for example Figure 9c which includes the three cycles also shown in Figure 4).

   We did include the temperature values since these are important for the onset of sea-ice growth. The salinity values are not that important for showing the general behavior. Since we do not use actual salinity values, we cannot compute the density with equation 1. Therefore, there are also no density values on the contour lines. We made sure

that the scale of the salinity axis is the same for each cycle, so they can be compared.

**Changes in manuscript:**
We will make changes in the caption and the main text to clarify this issue.

5. *Figure 5 (and others). Why did you plot thickness as a measure of polynya presence? The definition is concentration based. Panels b and e in Figure 5 are not that helpful. The thickness is the same every year. The shading indicates there is a constant polynya in panel e right? Where on the T-S curve is the active polynya. It should only be during Regime IV, i.e. while on the straight line between density 1027.8 and 1027.7? Actually, how can you say MKH is a polynya? It looks ice free the whole year? I'm very confused here.*

   **Author's reply:**
   The model does not determine sea-ice concentration. The sea-ice thickness is plotted in Figures 7-9 to show the difference between polynya and non-polynya periods. For consistency we also plotted the sea-ice thickness in Figure 5. Again for consistency, we included the shading. Furthermore, it also shows that the sea-ice thickness remains constant for these two cases (in Figure 5).

   The polynya definition we are using for this model is shown on pg. 10 l. 15. Following this definition, we can say that MKH is basically one long polynya period, with polynya formation every year. Every year there is a little bit of ice formation (about 10 cm), and due to the brine rejection of this sea-ice formation, the water column becomes unstable, mixing warm waters to the surface and melting the sea-ice. Case MKH can be compared with the 2-overturn cycle in Figure 4. At point D (in Figure 4) the polynya has formed. This point is not clearly visible in Fig. 5f, but it is located around (T, S) = (0.6, 34.7), around the 'kink' between the two straight lines.

   The results of case MKH are of course odd, but that is because we believe important physics are missing in this case. We do call it a polynya to be consistent with the other cases.

**Changes in manuscript:**
We will include a more extensive discussion of cases MKL and MKH where we will address the significance of Figures 5b and 5e, and why we do call it a polynya period.

6. *Figure 6. Can you indicate the actual polynya years in CESM here? Is it every year? Similarly for Figure 7.*

   **Author's reply:**
   The polynya years are addressed in the caption of Figure 6. The polynya years for Figures 7-9 are also clear from the shading in subfigures a and b.

   **Changes in manuscript:**
   The color codes corresponding to polynya years will be added in the captions for Figures 6-9.

7. *In the summary and discussion, the authors mention that this box model is slightly extended. This needs to be expanded. You could not replicate the results from Martinson as I understand it. More detail here on what is new about your study!*

   **Author's reply:**
   The main extension of the model is the inclusion of subsurface advection of heat and salt for both layers. Although we couldn't exactly replicate results from Martinson, we considered a case in which there is no subsurface advection (e.g. reference case). This set-up was the original set-up of Martinson and we found similar (not identical) results as reported in Martinson et al. (1981). All the other experiments have subsurface advection in the layers.

   **Changes in manuscript:**
   We will discuss and highlight the extensions of the original Martinson model in the discussion.

8. *Also, I sort of feel like it is missing a big punchline. What have you added to the body of literature on Maud Rise polynyas here? Is it just the enhanced role of subsurface heat accumulation? The results from the van Westen work were not simulated with the box model, but I think more needs to be added here to explain what the box model gives you and adds to the story.*

**Author's reply:**
Our starting point is the Martinson paper. In our view, this paper has a large influence on the general paradigm on Maud Rise polynya formation (i.e. deep convection induced by surface processes). What we have shown is that this model is not capable of simulating multiple polynya events as is seen in observations (e.g. the 1970s, 1980, 1994, and the 2016-17 event). When this model is extended, with most prominently (periodic) subsurface heat and salt accumulation, the model is capable of simulating multiple events. This is an improvement of the original model, which also sheds another light on the processes responsible for polynya formation.

Our model is very simple and includes only a few basic physical processes compared to that of the high-resolution CESM. Nevertheless, our model is capable of qualitatively reproducing the CESM simulation. This suggests that the most important physical processes are included in our model. The results suggest that subsurface heat and salt accumulation play an important role in polynya formation. Processes which have not been discussed often in the 'polynya literature'. Most studies only investigate surface instabilities (e.g. brine rejection) rather than subsurface processes. Surface forcing, which is also incorporated in the box model, is random and not causing polynya formation. Surface related processes cannot completely explain polynya formation nor its periodicity (if such a multidecadal period exists in the Southern Ocean, see discussion van Westen and Dijkstra (2020a)).

**Changes in manuscript:**
An extra paragraph will be added to highlight these findings and that the subsurface related processes need to be investigated in future research of the Maud Rise polynya.

References:

– Holland et al. (2014), Modeled Trends in Antarctic Sea Ice Thickness, https://doi.org/10.1175/JCLI-D-13-00301.1

– Trenberth et al. (2007). Estimates of the global water budget and its annual cycle using observational and model data, https://doi.org/10.1175/JHM600.1

– van Westen et al. (2020), Ocean model resolution dependence of Caribbean sea-level projections, https://doi.org/10.1038/s41598-020-71563-0

– van Westen and Dijkstra (2020a), Multidecadal Preconditioning of the Maud Rise Polynya Region, https://doi.org/10.5194/os-2020-25

– van Westen and Dijkstra (2020b), Subsurface Initiation of Deep Convection near Maud Rise, https://doi.org/10.5194/os-2020-33

**MS-No.:** OS-2020-63

**Title:** Multidecadal Polynya Formation in a Conceptual (Box) Model

**Author(s):** Daan Boot, René M. van Westen and Henk A. Dijkstra

**Point-by-point reply to reviewer #2**

**November 2, 2020**

We thank Wilbert Weijer for his careful reading and for the useful comments on the manuscript.

1. *p.1, l. 12: Usually a distinction is made between MRPs, which are clearly related to bathymetry; and the larger Weddell Sea Polynyas (WSPs) which are not related to bathymetry, as exemplified by those observed in the mid-70s. I suggest that the authors note this distinction.*

   **Author's reply:**
   Thank you for the notification. We will address this issue in the introduction.

   **Changes in manuscript:**
   The distinction between MRPs and WSPs will be made in the introduction.

2. *p.3, l. 2: vertical -> vertically stacked.*
   **Author's reply:**
   Suggestion followed.

   **Changes in manuscript:**
   The text will be changed accordingly.

3. *p.3, l. 27: remove 'a'*

   **Author's reply:**
   Suggestion followed.

   **Changes in manuscript:**
   The text will be changed accordingly.

4. *p. 5-6: In my opinion, the model description is adequate, maybe with the exception of the sea ice equation, which could use some clarification.*

   **Author's reply:**
   We agree, we will clarify the sea-ice equation.

   **Changes in manuscript:**
   The equations, including the sea ice equation, will be discussed more elaborately.

5. *p. 9, Caption Table 2: What do the 'bars' refer to? Overbars?*

   **Author's reply:**
   Yes, they refer to overbars.

   **Changes in manuscript:**
   The caption will be changed accordingly.

6. *p. 9, l. 21: So are these fluxes averaged over the polynya region?*

   **Author's reply:**
   Yes, they are spatially averaged over the 'polynya region' identified in Van Westen and Dijkstra (2020) (2°E – 11°E × 63.5°S – 66.5°S). This is mentioned in Figure 2.

   **Changes in manuscript:**
   The spatially averaging will also be mentioned in the main text, and the caption of Table 3.

7. *p. 10, l. 7: It would probably be good to explicitly state that this is a prescribed 25-yr cycle.*

   **Author's reply:**
   Suggestion followed.

   **Changes in manuscript:**
   It will be explicitly stated in the text that the forcing in the box model has a prescribed 25-year period.

8. *p. 10, l. 9: I think it would be good to have a better justification of the advective terms somewhere. A source of heat or salt is of course a consequence of a /divergence/ of advective fluxes. Maybe a better paradigm is that the lower box is âbathingâ in a water mass with ambient temperature $T_{b2}$ and salinity $S_{b2}$.*

**Author's reply:**
The reasoning behind the advective fluxes is that the layers do not drift away from the surrounding water masses. This reasoning follows your suggested paradigm. This has not been made explicit in the text. We will do that.

**Changes in manuscript:**
The justification of the advection terms will be changed to give a more physical point of view, as suggested in the above comment.

9. *p.10, l. 16: I suspect you mean ocean cooling, so heat transfer from the ocean to atmosphere. This would mean warming of the atmosphere.*

**Author's reply:**
Yes, that is what is meant here.

**Changes in manuscript:**
The statement in the text will be clarified.

10. *p. 11, Fig. 3: I'm a bit concerned by the strong variations, especially in the later years. I assume that the authors have checked that this water mass was not influenced by a polynya in the CESM? Evidently, you want to force the box model with upstream conditions.*

**Author's reply:**
We agree that forcing the model should be forced with upstream data. We therefore used a different region for fitting the subsurface fluxes (11°E – 12°E × 63.5°S – 66.5°S). We performed the model simulations with the new subsurface fluxes. The results become more convincing with the new fluxes. Especially case PFB improves with respect to the CESM simulation. Also the 25-year period becomes more apparent in the spectral analysis.

**Changes in manuscript:**
The new subsurface fluxes will be added, as well as the new results and a new discussion on the results.

11. *p. 15, l. 25-29: Maybe you can leave out the inclusion of the factor 35 (or discuss it somewhere else)? As it stands, $F_N$ is /not/ a freshwater flux, as claimed in l. 27, but a salt flux. Besides, it would result in a sign error.*

**Author's reply:**
Suggestion followed.

**Changes in manuscript:**
The text will be changed accordingly.

12. *p. 20-21: I think we are missing some rules for $T_2$ and $S_2$ in certain transitions.*

    **Author's reply:**
    You are right, those are missing.

    **Changes in manuscript:**
    The missing information will be included in the revision.

13. *In our recent paper (Kaufman et al. 2020) we studied the heat content in E3SMv0-HR (a close clone of CESM1), and also found that heat build-up preceded polynya formation. However, our analysis suggests that this heat build-up is driven by a reduced surface heat loss under ice-covered conditions, and not an enhanced ocean heat import (Fig. 8c). In fact, ocean heat advection appeared to counteract the heat accumulation by removing excess heat. I suppose that in the context of this box model, this situation would be represented by $T_2 > T_{b2}$ for long periods of time without polynyas. Does a situation like this occur in your model, and can you discuss the context of these occurrences?*

    *Reference: Kaufman, Z.S., Feldl, N., Weijer, W. and Veneziani, M., 2020. Causal Interactions between Southern Ocean Polynyas and High-Latitude Atmosphere Ocean Variability. Journal of Climate, 33(11), pp.4891-4905.*

    **Author's reply:**
    In our box model the situation that $T_2 > T_{b2}$ does not occur. The advective flux is thus always a source of heat to the subsurface layer. $T_2$ is always smaller than $T_{b2}$ because the subsurface layer loses heat to the surface layer via the term $K_T(T_1 - T_2)$.

    **Changes in manuscript:**
    This will be discussed in the revised manuscript.

References:

– van Westen and Dijkstra (2020), Multidecadal Preconditioning of the Maud Rise Polynya Region, https://doi.org/10.5194/os-2020-25

**MS-No.:** OS-2020-63

**Title:** Multidecadal Polynya Formation in a Conceptual (Box) Model

**Author(s):** Daan Boot, René M. van Westen and Henk A. Dijkstra

**Additional revisions as a reply to reviewer #2**

December 31, 2020

1. *First, the authors now make the distinction between MRPs and WSPs. I'm a bit surprised that the authors present the box model as model for MRPs, whereas in my mind it is more appropriately applied to WSPs. WSPs are thought to be associated with convection due to large-scale instabilities of the water column, as modeled here. In contrast, MRPs appear to have a significant dynamic driver, namely the Taylor cap dynamics. It would be good to understand why the authors apply this model to MRPs, but not to WSPs. Besides, Martinson et al., Dufour et al. all consider WSPs.*

   **Author's reply:**
   As mentioned by the reviewer, stratified Taylor columns (Alverson and Owens, 1996, de Steur et al., 2007) contribute to the preconditioning of the Maud Rise region. There are still two distinct layers near Maud Rise in the presence of stratified Taylor columns (see Figure 3a). Taylor columns only reduce the stratification near Maud Rise and make this region more susceptible to convection compared to the surroundings. However, the stratified Taylor columns near Maud Rise are not sufficient to initiate convection (van Westen and Dijkstra 2020a). Most MRP literature (Kurtakoti et al., 2018; Campbell et al., 2019; Cheon and Gordon, 2019; Kaufman et al., 2020) consider positive salinity anomalies to initiate the convection near Maud Rise, similar as in WSP formation in the model of Martinson et al. (1981). This suggests that the Martinson model can still be used, but needs some adjustment for the Maud Rise region. For example, the total depth ($H = 2000$ m) is adjusted and we use different values for the subsurface temperature ($T_2$) and salinity ($S_2$) compared to Martinson et al. (1981). The effect of stratified Taylor columns are assimilated in the subsurface temperature and salinity time series (Figure 4). These values are retained from

model output of the Community Earth System Model (CESM). In this way, the basic set-up of the Martinson model can be used to study MRP formation.

**Changes in manuscript:**
The differences between this study (MRPs) and Martinson et al. (1981) are discussed in the revision. We motivate why the original model is still applicable, with some adjustments, to the Maud Rise region (pg. 2 lines 32, 33; pg. 3 lines 1-6).

2. *Also, there are still a few issues with the set of equations. For starters, Eq. 5b is missing the exchange term with the layer below. Second, there is still a problem with the freshwater flux. I now realize that my previous comment about F was ignoring a bigger issue with how the term F is treated and described; I realize that Martinson et al. (1981) are treating this term rather loosely, but it would be good to be consistent here. The simplest fix is to simply replace F by $F S_0$ in Eqs. 2b, 3b, 4b and 5b; as well as in 4c and 5c. Martinson et al. use a conversion factor of 35 g/kg. In that way, F still represent the freshwater flux (positive if into the ocean), with units of m/s (or similar); while $F S_0$ represents the associated virtual salt flux. Please verify that the code is correct, and that these issues are addressed in the manuscript.*

**Author's reply:**
This is not clearly documented in the revision and we follow the suggestion by the reviewer. The freshwater flux has been used correctly in the model code, i.e. as a virtual salt flux.

**Changes in manuscript:**
The term is added in equation 5b. $S_0$ is added to the equations, the discussion of the equations, and to Table 2. Transforming the freshwater flux to a virtual salt flux is also discussed in section 2.1.

3. *In the sections on p. 6 please be more consistent in the terminology: l. 12, ocean atmosphere heat flux is given by $Q_{ia}$ –without the terms in the denominator, which convert the flux to a temperature change. Similarly in ll. 15, 17, 19, 22, 24, and 25.*

**Author's reply:**
The terminology was confusing and we aligned the terminology throughout the manuscript.

**Changes in manuscript:**
The terminology is aligned throughout the manuscript.

List of all relevant changes:

- The subsurface fluxes have been fitted to a different region. This means we have used different subsurface fluxes. The description, figures and discussion have been revised. The results have not changed qualitatively and have become more convincing. The data sets made available have also been updated.

- In Section 2 a paragraph has been added regarding the CESM simulation used for this study. Polynya formation in the simulation is also addressed in this paragraph.

- The discussion of the model equations is more elaborate.

- Section 4 (Summary and discussion) is extended with an extra paragraph highlighting our findings.

- The convention h, and H-h for the layer dephts has been changed to $h_1$ and $h_2$.

- A distinction between Weddell Sea Polynyas and Maud Rise Polynyas is made.

- In the original manuscript the freshwater flux was not treated consistently. It has been made clear that the freshwater flux is converted to a virtual salt flux in the model.

- Other changes are relatively minor and mostly small textual changes.